# Genetic characteristics of human parainfluenza viruses 1–4 associated with acute lower respiratory tract infection in Chinese children, during 2015–2021

Yun Zhu,[1,2] Yun Sun,[3] Changchong Li,[4] Gen Lu,[5] Rong Jin,[6] Baoping Xu,[7] Yunxiao Shang,[8] Junhong Ai,[1,2] Ran Wang,[1,2] Yali Duan,[1,2] Xiangpeng Chen,[1,2] Zhengde Xie[1,2]

**ABSTRACT**   Human parainfluenza viruses (HPIVs) are a significant cause of acute lower respiratory tract infections (ALRTIs) among young children and elderly individuals worldwide. The four types of HPIVs (HPIV1–4) can cause recurrent infections and pose a significant economic burden on health care systems globally. However, owing to the limited availability of complete genome sequences, the genetic evolution of these viruses and the development of vaccines and antiviral treatments are hampered. To address this issue, this study utilized next-generation sequencing to obtain 156 complete genome sequences of HPIV1–4, which were isolated from hospitalized children with ALRTIs in six regions of China between 2015 and 2021. This study revealed multiple clades, lineages, or sublineages of HPIVs circulating in mainland China, with a novel clade D of HPIV1 identified as geographically restricted to China. Moreover, this study identified the endemic dominant genotype of HPIV3, lineage C3, which has widely spread and continuously circulated in China. Bioinformatic analysis of the genome sequences revealed that the proteins of HPIV3 possessed the most variable sites, with the P protein showing more diversity than the other proteins among all types of HPIVs. The HN proteins of HPIV1–3 are all under negative/purifying selection, and two amino acid substitutions in the HN proteins correspond to known mAb neutralizing sites in the two HPIV3 strains. These findings provide crucial insights into the genetic diversity and evolutionary dynamics of HPIVs circulating among children in China and may facilitate research on the molecular diagnosis, vaccine development, and surveillance of HPIVs.

**IMPORTANCE**   Phylogenetic analysis revealed the prevalence of multiple clades, lineages, or sublineages of human parainfluenza viruses (HPIVs) circulating in mainland China. Notably, a unique evolutionary branch of HPIV1 containing only Chinese strains was identified and designated clade D. Furthermore, in 2023, HPIV3 strains from Pakistan and Russia formed a new lineage within clade C, named C6. The first HPIV4b sequence obtained in this study from China belongs to lineage C2. Evolutionary rate assessments revealed that both the HN and whole-genome sequences of HPIV3 presented the lowest evolutionary rates compared with those of the other HPIV types, with rates of 6.98E−04 substitutions/site/year (95% HPD: 5.87E−04 to 8.25E−03) and 5.85E−04 substitutions/site/year (95% HPD: 5.12E−04 to 6.62E−04), respectively. Recombination analysis revealed a potential recombination event in the F gene of an HPIV1 strain in this study. Additionally, all the newly obtained HPIV1–3 strains exhibited negative selection pressure, and two mutations were identified in the HN protein of two HPIV3 strains at monoclonal antibody-binding sites.

**KEYWORDS**   human parainfluenza virus, genetic characteristics, children, acute lower respiratory tract infection, reassortment

Address correspondence to Zhengde Xie, xiezhengde@bch.com.cn, or Xiangpeng Chen, chenxp1111@163.com.

The authors declare no conflict of interest.

See the funding table on p. 16.

Human parainfluenza viruses (HPIVs) are a diverse group of enveloped, negative-sense RNA viruses belonging to the *Paramyxoviridae* family (1). These viruses are classified into four types, namely, HPIV1, HPIV2, HPIV3, and HPIV4, based on their genetic and antigenic characteristics (1). HPIV1 and HPIV3 are classified into the *Respirovirus* genus, whereas HPIV2 and HPIV4 are classified into the *Rubulavirus* genus (1–3). Additionally, HPIV4 can be further classified into two subtypes, 4a and 4b, based on hemagglutination inhibition and neutralizing tests (4, 5). The genomes of HPIVs range from 15,000 to 17,000 nucleotides and encode six main structural proteins, including nucleocapsid protein (N), phosphoprotein (P), matrix protein (M), fusion (F), hemagglutinin-neuraminidase (HN), and large protein (L) (1, 6–8). The HN and F surface glycoproteins mediate host cell receptor binding and facilitate viral entry. In addition, they represent key targets of the specific host immune response (9, 10).

HPIVs significantly cause acute lower respiratory tract infections (ALRTIs) in children under 5 years of age (11, 12), contributing to 4%–14% of ALRTI hospital admissions and 4% of childhood ALRTI-related deaths globally (11). HPIV infections are also associated with upper and lower respiratory tract illnesses, including the common cold, croup, tracheobronchitis, bronchiolitis, and pneumonia, in both children and adults (13, 14). Although these infections are typically mild in healthy individuals, they may lead to severe respiratory diseases in children and immunocompromised individuals (15, 16). Individuals are susceptible to recurrent HPIV infections throughout their life owing to incomplete protective immunity to these viruses (17). Serological studies have shown that up to 80% of children are infected with HPIV3 by the age of four (18–20).

Currently, no licensed vaccine or antiviral agent is available to prevent or treat HPIV infections (21, 22). Further genetic data on HPIVs are crucial for the development of vaccines and antiviral agents. However, publicly available HPIV1–4 sequences are limited, especially from China. To fill this gap, we conducted a study involving whole-genome sequencing and analysis of HPIV1–4 strains collected from hospitalized children with ALRTI in mainland China from 2015 to 2021. Our findings contribute to a broader understanding of the genetic diversity of HPIVs and advance the development of safe and effective vaccines and antiviral agents against these viruses.

## MATERIALS AND METHODS

### Patients and specimens

From 2015 to 2021, respiratory samples, such as nasopharyngeal swabs and nasopharyngeal aspirates, were obtained from hospitalized children with ALRTI at six children's hospitals in China, including Beijing Children's Hospital, Capital Medical University in Beijing, Shengjing Hospital of China Medical University in Liaoning Province, Yinchuan Women and Children Health care Hospital in Ningxia Hui Autonomous Region, The 2nd Affiliated Hospital and Yuying Children's Hospital of Wenzhou Medical University in Zhejiang Province, Guangzhou Women and Children's Medical Center in Guangdong Province, and Guizhou Maternal and Child Health Care Hospital. All samples were collected within 24 hours of admission, preserved in viral transport media, and transported to Beijing Children's Hospital using dry ice. Upon receipt at Beijing Children's Hospital, the samples were stored at −80℃ until further processing. Demographic, epidemiologic, and clinical data were documented by clinical researchers via a uniform case report form.

### RNA extraction and HPIV1–4 detection

Viral RNA was extracted from each sample via a QIAamp MinElute Virus Spin Kit (Qiagen, Germany) following the manufacturer's instructions. HPIV1–4 and other respiratory viruses were screened with Luminex RVP Fast V2 kits via Luminex Magpix (Luminex, USA) according to established protocols (12).

## Sequencing complete genomes of HPIV1–4

All the HPIV-positive samples were further confirmed via a Real-time reverse-transcription polymerase chain reaction kit for HPIV1–4 (BioGerm, Shanghai, China). The samples with Ct values of less than 28 were subsequently sent to the Shanghai BioGerm Medical Technology Limited Company for next-generation sequencing (NGS). The nucleic acids extracted from the HPIV-positive samples were captured using an HPIV genome enrichment kit from Shanghai BioGerm Medical Technology Co., Ltd (BioGerm). A series of overlapping PCR amplicons were generated using primers designed by BioGerm to efficiently sequence viral genomes directly from clinical specimens. The amplified PCR product was purified and quantified, after which an Illumina Nextera XT Kit was used for deep sequencing via Illumina MiSeq. The sequencing results were analyzed via the CLC Genomics Workbench 12 (Qiagen, Germany). More than 90% of the sequencing reads reached Q30 (99.9% base call accuracy). The sequencing data volume of each sample was 1 Gb, with 22–33 million reads. A sequencing depth of more than 8,000× was used for mapping to the reference, with a comparison rate of greater than 99.99%.

## Data set

A total of 978, 695, 3221, and 411 HPIV1–4 nucleotide sequences publicly available as of 31 December 2023 were retrieved from GenBank. Incomplete hemagglutinin-neuraminidase (HN) or whole-genome sequences, clone sequences, modified microbial nucleic acids, synthetic constructs, strains passaged in culture multiple times, or sequences containing ambiguous nucleotides were excluded from the analysis. To ensure representativeness and reduce redundancy, one nonidentical sequence per epidemic season/year and country was chosen for further analysis. A data set was assembled for analysis, consisting of 94, 120, 152, and 81 complete HN coding sequence (CDS) sequences of HPIV1–4, along with 61, 61, 120, and 46 whole-genome sequences (WGSs) of HPIV1–4, which were collected globally from GenBank from 1955 to 2023 (shown in Table S2). Additionally, 45, 9, 101, and 1 sequences of HPIV1–4 collected from six provinces in this study were included for further analysis. Multiple sequence alignment was conducted using MAFFT version 7.471 software.

## Nucleotide identity, phylogenetic analysis, and evolutionary rate

The nucleotide identity among each type of HPIV1–4 sequence obtained in this study was measured via BioEdit software version 7.1.3.0. These strains were compared with the prototype strain in terms of nucleotide identity. The genetic distance (p-distance) within and between each clade/lineage/sublineage was assessed using MEGA version 7.0.26. Owing to the large size of the HPIV3 data set, which includes more than 200 sequences for both HN (252 sequences) and WGS (221 sequences), we initially constructed maximum likelihood (ML) phylogenetic trees for HPIV3 based on HN and WGS sequences, respectively. On the basis of the ML tree of the HN and WGS sequences, we selected strains obtained in this study from the same evolutionary branch, matching according to the year of collection and geographical location. Ultimately, 20 representative strains sequenced in this study were chosen with reference strains for Bayesian phylogenetic tree inference, evolutionary rate estimation, and genetic distance analysis. Additionally, we constructed maximum likelihood (ML) phylogenetic trees for the HN and for the WGS of other HPIV types. The ML phylogenetic trees were generated using MEGA software version 7.0.26, which employs the Kimura 2-parameter model of nucleotide substitution. The reliability of the tree topology was assessed through 1,000 bootstrap replicates, with bootstrap values exceeding 70 considered strong support for the branching order. Bayesian phylogenetic trees for the HN gene and WGS of HPIV1–4 were constructed via the Bayesian Markov chain Monte Carlo (MCMC) method implemented in BEAST v1.10.4 software (Bouckaert et al., 2014). The appropriate substitution model (GTR + G + I) was determined using jModelTest 2.1.10 (Darriba et al., 2012). The uncorrelated relaxed molecular clock model, along with the constant size as the coalescent tree

prior, was used as the parameter of the Bayesian phylogenetic models. Convergence of the data set was assessed using Tracer v1.7.2 software, with an MCMC chain length of 300,000,000 steps and sampling every 10,000 steps. Convergence was confirmed by evaluating effective sample sizes (ESSs) in Tracer v1.10.4, accepting values greater than 200. The maximum clade credibility trees were generated after discarding 10% burn-in trees via TreeAnnotator v1.10.4 within BEAST v1.10.4 software. The resulting Bayesian MCMC phylogenetic tree phylogenetic classification of the HN gene of HPIV1–4 was based on the topology of the Bayesian MCMC phylogenetic tree, with clades, lineages, and sublineages named according to established criteria of genetic distances and previous studies (3, 7, 23–25) with branch reliability supported by 95% highest posterior density (HPD). The classification of HPIV1–4 WGSs followed similar criteria as those applied for the HN gene. All the trees were visualized and annotated using FigTree v1.4.3 (http://tree.bio.ed.ac.uk/software/figtree/) and Adobe illustrator CC 2019. The rates of molecular evolution of HPIV1–4 were estimated. The 95% HPD distribution was used to describe the 95% confidence intervals of the analysis results.

## Recombination analyses

Recombination analyses of HPIV1–4 WGS sequences were conducted via various algorithms, including RDP, GNECONV, BootScan, MaxChi, Chimaera, SiSan, and 3Seq, within the RDP4 software (26). To ensure the validity of the results, recombination events were considered genuine only if they were detected by at least three of the algorithms. The putative recombination sequence was subsequently subjected to additional confirmation and analysis via Simplot software, which employs similarity and bootscanning analyses with a sliding window size of 200 bp and a moving step size of 20 bp.

## Positive and negative selection site analysis

Selective pressure on site was assessed via the ratio of the number of nonsynonymous substitutions per nonsynonymous (dN) to that of synonymous substitutions per synonymous site (dS), employing multiple algorithms, including SLAC, FEL, MEM, and FUBAR, available via Datamonkey online software (http://www.datamonkey.org/) (27). The potential positive (PSS) and negative (NSS) selected sites on the predicted HN and F proteins of HPIV1–3 obtained in this study were evaluated. Sites were classified as positive if they met the criteria of at least two algorithms: a $P$ value of less than 0.05 for SLAC, FEL, and MEME, and a Bayes factor/posterior probability of greater than 0.95 for FUBAR (7, 27).

## Glycosylation site analysis

The predicted amino acid sequences of the F and HN genes of HPIV1–4 were utilized to predict N-glycosylation (https://services.healthtech.dtu.dk/services/NetNGlyc-1.0/) and GalNAc O-glycosylation (https://services.healthtech.dtu.dk/services/NetOGlyc-4.0/) site prediction via the NetNGlyc 1.0 and 4.0 servers, respectively. The glycosylated sites were identified using a threshold score of 0.5 or higher.

## Entropy plots

To identify regions with high mutation rates in the CDSs of the N, P, M, F, HN, and L genes of HPIV1–4, the complete genome alignments were first trimmed to include only the CDS regions and subsequently translated into predicted protein sequences. The entropy information was calculated for all the positions via the Entropy (H(x)) plot function available in BioEdit software version 7.1.3.0 (7). The entropy values for the CDS of HPIV1–4 were plotted using Microsoft Excel. The variable site was defined as a position with variants observed in more than one sequence, and the percentage of variable sites for each protein was determined.

## Nucleotide sequence accession numbers

The nucleotide sequences obtained in this study are deposited in the GenBank database under accession numbers OQ981667–OQ981711 for HPIV1, OQ990765–OQ990773 for HPIV2, OQ981712–OQ981812 for HPIV3, and OQ990818 for HPIV4 and are shown in Table S1.

## RESULTS

### Sequence identity analysis

A total of 156 complete genome sequences of HPIVs were obtained from clinical samples, comprising 45 HPIV1, 9 HPIV2, 101 HPIV3, and 1 HPIV4b sequences. The basic clinical information for all these sequences is summarized in Table S1. For the HPIV1–4 sequences obtained in this study, among the 45 HPIV1 sequences, nucleotide identity ranged from 96.2% to 100% between these sequences and from 95.1% to 95.6% compared with the prototype strain (AF457102 HPIV1/USA/1964). For the 9 HPIV2 sequences, the nucleotide identity ranged from 93.1% to 99.5% between these sequences and from 94.6% to 96.9% compared with that of the prototype strain (AF533012_HPIV2/USA/GREER/1955). Among the 101 HPIV3 sequences, nucleotide identity ranged from 97.3% to 100% (the complete genome sequence of the HPIV3 prototype strain was unavailable). The nucleotide similarity between the HPIV4b sequence in this study and the prototype strain (AB543337_HPIV4b/JPN/Tokyo_68–333/1968) was 94.6%.

### Phylogenetic analysis of the HN CDS and WGS data for HPIV1–4

Phylogenetic trees based on the HN CDS of HPIV1–4 were constructed (Fig. 1; Fig. S1). For the HN CDS of HPIV1, the Bayesian phylogenetic tree was categorized into four distinct clades: A, B, C, and D. Interestingly, the eight HPIV1 strains from Beijing and Zhejiang Provinces sequenced in this study formed a new clade, designated clade D, along with a strain detected in Jilin Province, China. The genetic distance between these four clades of HPIV1 ranged from 0.020 ± 0.003 (between clades A and C) to 0.042 ± 0.004 (between clades C and D). The newly defined clade D exhibited genetic distances from other clades as follows: 0.028 ± 0.004 (between clades A and D), 0.027 ± 0.003 (between clades B and D), and 0.042 ± 0.004 (between clades C and D). Therefore, we designated this cluster clade D (Fig. S3A). The analysis of the HN gene of HPIV2 via CDS identified four clades (G1–G4) with additional lineages: lineages G1a, G1b, and G1c in clade G1 and lineages G4a and G4b in clade G4 (Fig. 1B; Fig. S3B). Before this study, several HPIV2 strains had been reported in China, falling into lineages G1a and G1c and clade G3. The viruses sequenced in this study belonged to clades G3 (n = 3) and lineage G1a (n = 6) (Fig. 1B) and circulated with strains from North America, Asia, and Europe. For HPIV3, the ML trees of the HN gene clustered into three clades (A, B, and C), with clade C further subdivided into lineages C1–C6. Among these, lineage C1 could be further classified into sublineage C1a–C1d, and lineage C3 showed the most diversity and could be divided into sublineages C3a–C3g, with genetic distances ranging from 0.009 ± 0.001 to 0.023 ± 0.003. All HPIV3 strains detected in China were clustered into sublineages C3a, C3b, C3c, C3e, and C3f, except for one strain from Zhejiang Province in 2018, which belonged to the sublineage C1c. The HPIV3 viruses sequenced in this study belong to sublineages C3a, C3b, and C3f (Fig. 1C; Fig. S1C), which have clustered together with strains circulating in Asia, North America, South America, Europe, and Africa since 2000. Furthermore, our study identified a new lineage, designated C6, formed by strains from Pakistan and Russia in 2023. The genetic distances between lineage C6 and other lineages within clade C range from 0.043 ± 0.005 (between lineages C3 and C4) to 0.060 ± 0.005 (between lineages C5 and C6), which are considerably greater than the minimum genetic distance of 0.022 ± 0.002 observed between lineages C3 and C5 previously reported within clade C (Fig. S3C). For HPIV4, the analysis based on the CDS of the HN gene clearly revealed two distinct clusters, 4a and 4b. Within both subtypes

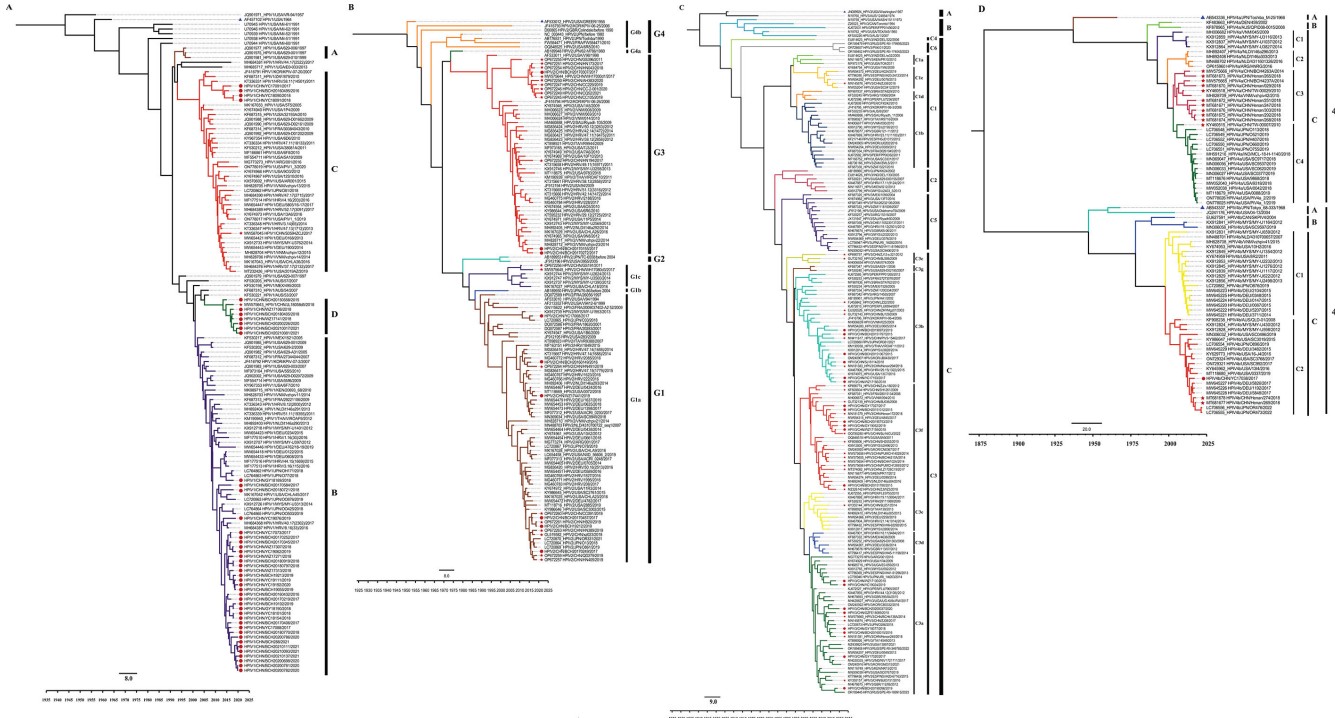

**FIG 1** Bayesian phylogenetic tree based on the full-length HN CDS sequence of HPIV1–4. Fig. (A–D) correspond to the trees of HPIV1, HPIV2, HPIV3, and HPIV4, respectively. The prototype strains, the strains obtained in this study, and other Chinese strains are indicated by blue triangles, red solid circles, and red pentagrams, respectively. The names of the strains include the GenBank number, serotype, country of isolation, name, and year of the collection. The country abbreviations ARG, AUS, BRA, CHE, CHN, DEU, ESP, FRA, GBR, HRV, ITA, IND, JPN, KEN, KOR, MEX, MYS, NLD, PER, RUS, SAU, THA, UGA, USA, VNM, ZAF, and ZAM in the trees represent Argentina, Australia, Brazil, Switzerland, China, Germany, Spain, France, the United Kingdom, Croatia, Italy, India, Japan, Kenya, South Korea, Mexico, Malaysia, the Netherlands, Peru, Russia, Saudi Arabia, Thailand, the Republic of Uganda, the United States, Vietnam, the Republic of South Africa, and the Republic of Zambia, respectively.

4a and 4b, three primary evolutionary lineages have emerged, designated clades A, B, and C. Additionally, the clades within subtypes 4a and 4b can be further subdivided into lineages C1–C4 and C1–C2, respectively (Fig. 1D; Fig. S3D). Among these, a newly sequenced HPIV4b strain, HPIV4b/CHN/YC17038/2017, was found to belong to lineage C2 of subtype 4b, with close proximity to strains isolated from various geographical locations, including Japan, Germany, China, the United States, and India (Fig. 1D; Fig. S1D).

We also conducted Bayesian phylogenetic analysis via WGS of HPIV1–4. The Bayesian phylogenetic trees of the complete genome sequences of HPIV1–4 presented similar topologies that were all consistent with those of the corresponding type of HN gene. Nevertheless, certain clades, lineages, or sublineages can be identified only through their HN gene sequences. Consequently, these specific groups were not represented in the phylogenetic tree constructed based on the complete genome (Fig. 2A through D). Additionally, we constructed ML phylogenetic trees for HPIV1–4 based on HN and WGS. These ML trees exhibited similar topologies to those of the Bayesian phylogenetic trees, with consistent assignment of sequences to clades, lineages, and sublineages (Fig. S2A through D). This convergence provides robust support for the results obtained from the Bayesian phylogenetic tree analysis.

## Evolutionary rate of the HN CDS of HPIV1–4

The Bayesian analysis revealed the mean evolutionary rates of global HPIV1–4 HN CDSs, ranked from highest to lowest, as follows: 9.05E−04 substitutions/site/year (95% highest posterior density [HPD]: 7.65E−04 to 1.05E−03) for HPIV1, 8.12E−04 substitutions/site/

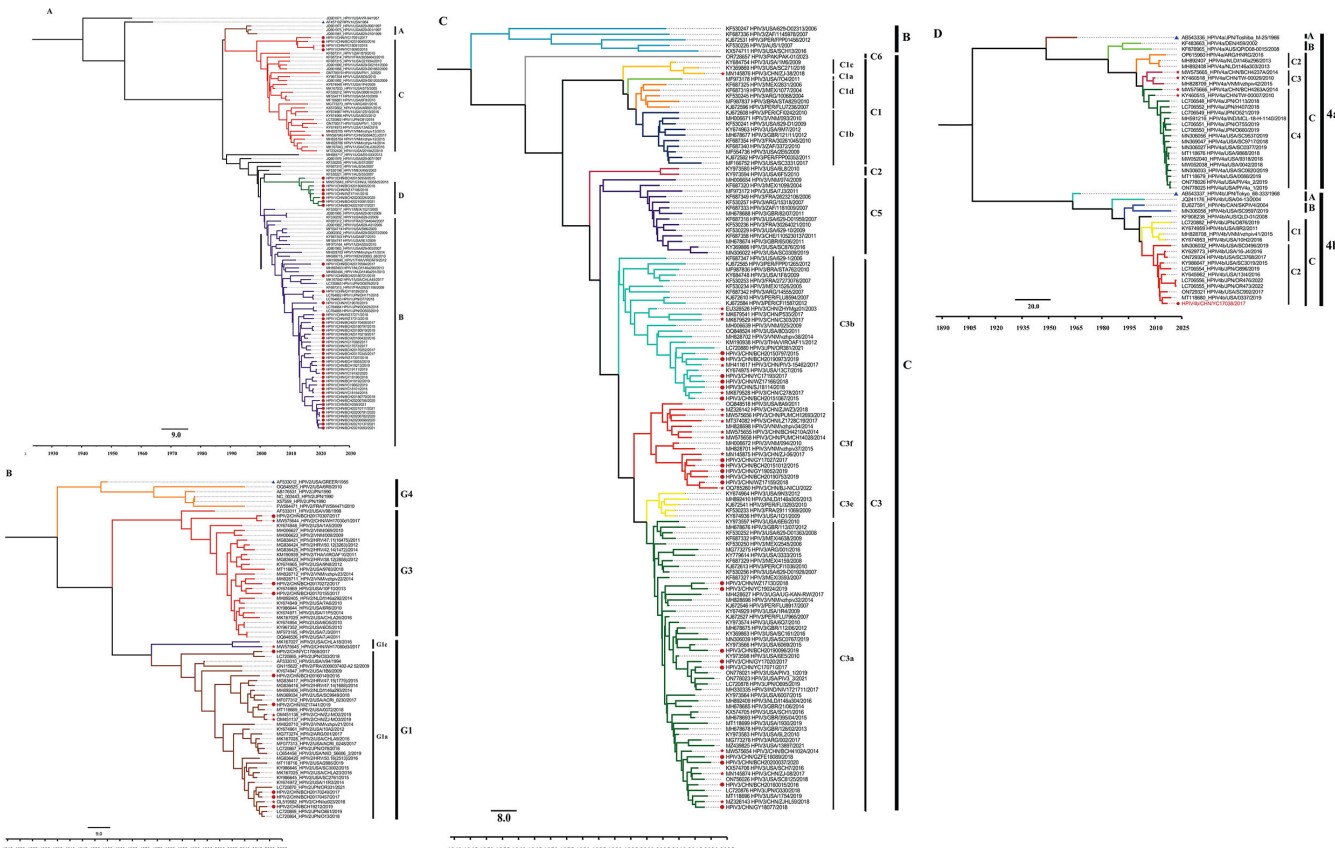

**FIG 2** Bayesian phylogenetic tree based on whole genome sequence of HPIV1–4. Fig. (A–D) correspond to the trees of HPIV1, HPIV2, HPIV3, and HPIV4, respectively. The prototype strains, the strains obtained in this study, and other Chinese strains are indicated by blue triangles, red solid circles, and red pentagrams, respectively. The names of the strains include the GenBank number, serotype, country of isolation, name, and year of the collection. The country abbreviations ARG, AUS, BRA, CHE, CHN, FRA, GBR, HRV, ITA, JPN, MEX, NLD, PER, THA, UGA, USA, and VNM in the trees represent Argentina, Australia, Brazil, Switzerland, China, France, the United Kingdom, Croatia, Italy, Japan, Mexico, the Netherlands, Peru, Thailand, the Republic of Uganda, the United States, and Vietnam, respectively.

year (95% HPD: 5.80E−04 to 1.09E−03) for HPIV2, 7.89E−04 substitutions/site/year (95% HPD: 5.86E−04 to 1.58E−03) for HPIV4, and 6.98E−04 substitutions/site/year (95% HPD: 5.87E−04 to 8.25E−03) for HPIV3. Furthermore, the Bayesian analysis estimated the mean evolutionary rates of global HPIV1–4 WGS, arranged from highest to lowest, as follows: 1.12E−03 substitutions/site/year (95% HPD: 7.77E−04 to 1.58E−03) for HPIV4, 7.11E−04 substitutions/site/year (95% HPD: 5.97E−04 to 8.37E−04) for HPIV1, 6.46E−04 substitutions/site/year (95% HPD: 3.20E−04 to 1.12E−03) for HPIV2, and 5.85E−04 substitutions/site/year (95% HPD: 5.12E−04 to 6.62E−04) for HPIV3 (Fig. 3).

## Recombination analysis

The aligned complete genomes of HPIV1–4 were analyzed to identify potential recombination events using RDP4 and Simplot software. The findings revealed that no HPIV2, HPIV3, or HPIV4b strains sequenced in this study demonstrated evidence of recombination. However, one HPIV1 strain (HPIV1/CHN/YC17073/2017, YC17073), which belongs to clade B, was identified to have undergone a recombination event. The potential major parent was identified as HPIV1/CHN/BCH20170252/2017 (clade B), which shares 99.5% nucleotide identity with YC17073. Additionally, the potential minor parent was identified as HPIV1/CHN/YC17091/2017 (clade C), which presented 96.4% nucleotide similarity with YC17073. The recombination event was found to have occurred between

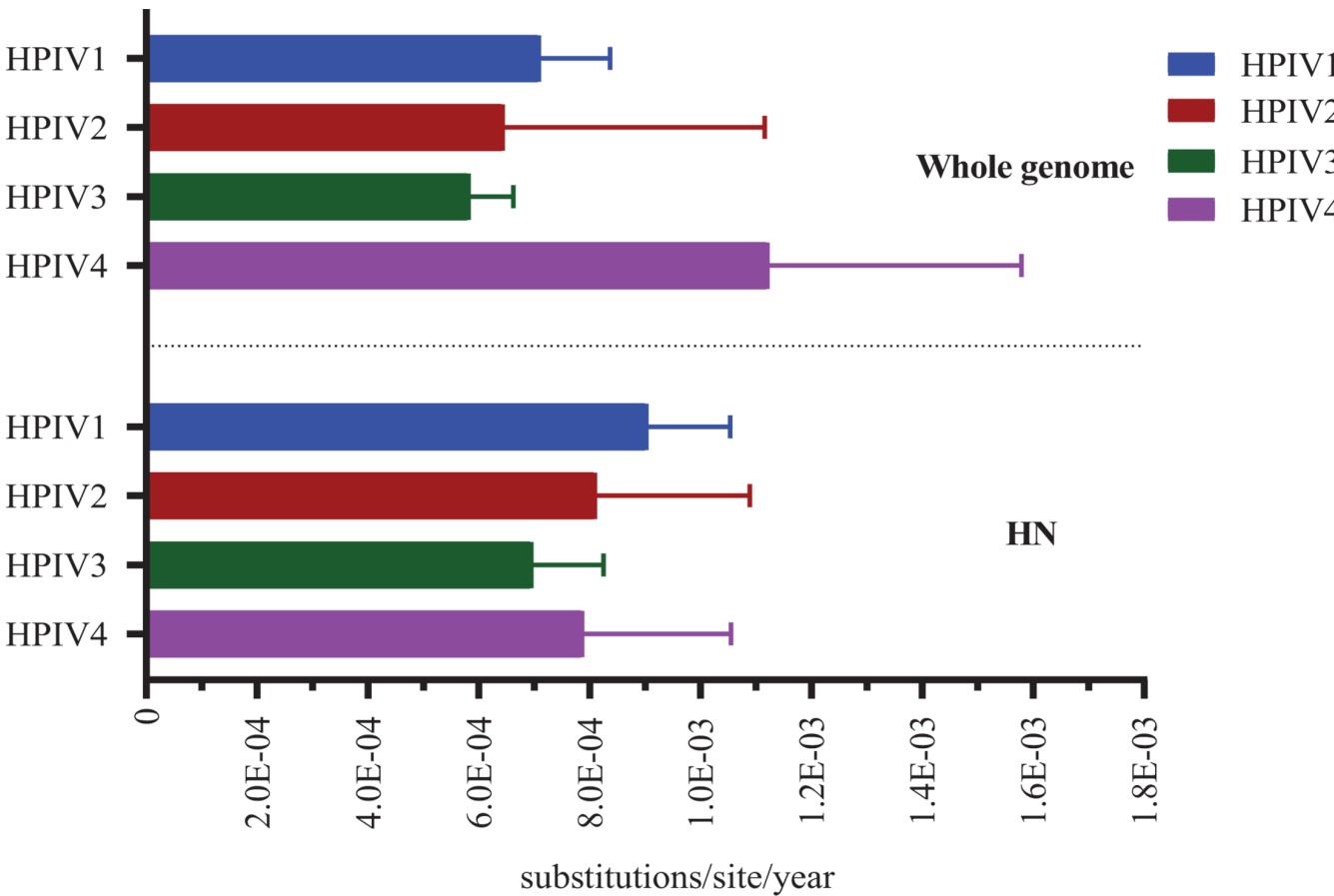

**FIG 3** Evolutionary rate of HN CDS and whole-genome sequences for globe HPIV1–4.

nucleotide positions 4497 and 5543, corresponding to the F gene region. This finding was further confirmed by simple and bootscan analyses via Simplot software (Fig. 4).

## Selective pressure analysis

The computation and assessment of evolutionary pressure is an essential component of a statistical toolbox used in sequence analysis to understand genetic variation. Selection pressure analysis revealed the absence of a positively selected site (PSS) in the HN and F genes of HPIV1–3 sequenced in this study. However, multiple negatively selected sites (NSSs) were identified by at least two of the four algorithms (SLAC, FLE, MEM, and FUBAR) (Table 1; Tables S3 and S4). For 45 newly sequenced HPIV1 strains, 11 NSSs in the HN protein and 10 NSSs in the F protein of clade B strains were detected. However, no NSSs were identified in the HN or F proteins of clade D or C strains. Among the nine HPIV2 strains, six NSSs in the HN protein and five NSSs in the F protein were inferred in clade G3. Additionally, only one NSS was found in the F protein of strains belonging to lineage G1a. Among the HPIV3 strains of sublineage C3a, 15 NSSs were found in the HN protein, and 23 NSSs were found in the F protein. Among those of sublineage C3b, we confirmed six NSSs in the HN protein and four NSSs in the F protein. For those of sublineage C3f, 21 NSSs in the HN protein and 32 NSSs in the F protein were inferred.

## Glycosylation site analysis

We conducted analyses of the N-glycosylation and O-glycosylation sites of 156 newly sequenced HPIV strains (Table S5). For 45 HPIV1 strains, the HN protein contained 6 conserved N-glycosylation sites. In addition, one strain presented an additional N-glycosylation site at amino acid position 77, and four clade C strains presented an extra

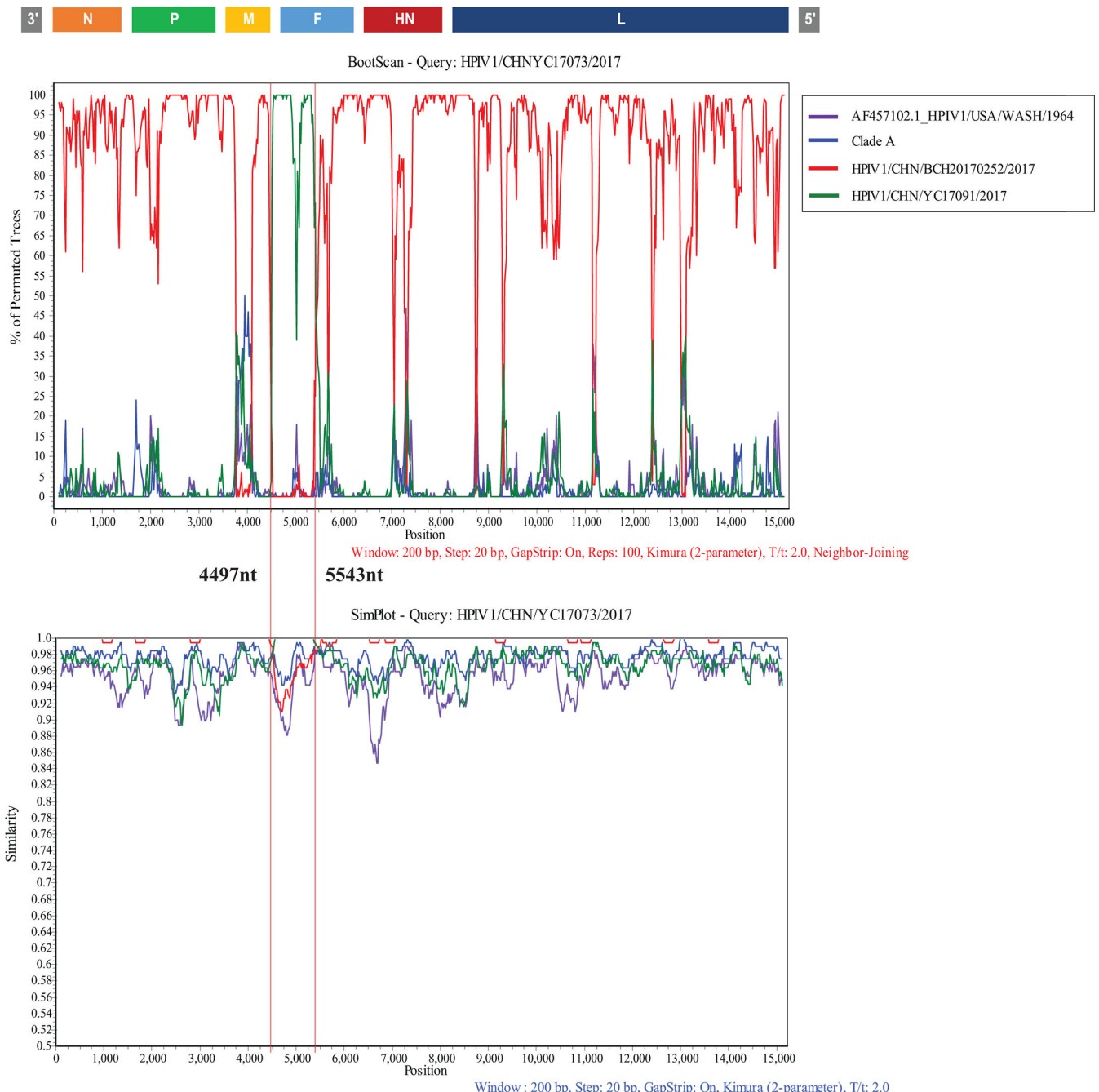

**FIG 4** Similarity and bootscaning analyses for complete CDS region of HPIV1 strain. Similarity plot (A) and bootscanning analysis (B) were performed using the full-length genome sequences of prototype strain HPIV1/USA Washington/1964 (GenBank accession: AF457102), grouped strains belonged to clade A, the potential major parent (HPIV1/China_Beijing/BCH20170252/2017) and the putative minor parent (HPIV1/China_Nixiang/YC17091/2017). A sliding window of 1,000 nucleotides (nt) moving in 200 nt step was used in this analysis.

N-glycosylation site at position 511. O-Glycosylation was commonly found at position 151 in the HN protein, and a few strains had predicted O-glycosylation sites at positions 74, 79, and 147. However, one strain, HPIV1/CHN/BCH20180721/2018, did not possess any O-glycosylation site in its HN protein. With respect to the F protein, a potential N-glycosylation site was inferred at amino acid position 241, but only a few strains presented N-glycosylation sites at positions 100 or 529. Similarly, O-glycosylation sites were commonly found at positions 102, 103, and 453 in most sequences, whereas a few

**TABLE 1** Selective pressure sites in the HN and F protein of HPIV1–3 obtained in this study[a]

| Virus | Clade/lineage/ sub-lineage | Viral protein | SLAC | | FUBAR | | FEL | | MEME |
|---|---|---|---|---|---|---|---|---|---|
| | | | PSS | NSS | PSS | NSS | PSS | NSS | PSS |
| HPIV1 | B | | 0 | 1 | 1 | 14 | 0 | 11 | 0 |
| | C | | 0 | 0 | 0 | 0 | 0 | 1 | 0 |
| | D | | 0 | 0 | 0 | 0 | 0 | 0 | 0 |
| HPIV2 | G3 | HN | 0 | 0 | 0 | 7 | 0 | 8 | 0 |
| | G1a | | 0 | 0 | 0 | 0 | 0 | 9 | 0 |
| HPIV3 | 3a | | 0 | 4 | 0 | 23 | 0 | 15 | 0 |
| | 3b | | 0 | 1 | 0 | 9 | 0 | 7 | 0 |
| | 3f | | 0 | 8 | 0 | 28 | 0 | 21 | 0 |
| HPIV1 | B | | 0 | 2 | 0 | 17 | 0 | 10 | 0 |
| | C | | 0 | 0 | 0 | 0 | 0 | 2 | 0 |
| | D | | 0 | 0 | 0 | 0 | 0 | 1 | 0 |
| HPIV2 | G3 | F | 0 | 0 | 0 | 5 | 0 | 6 | 0 |
| | G1a | | 0 | 0 | 1 | 1 | 0 | 5 | 0 |
| HPIV3 | 3a | | 0 | 6 | 0 | 32 | 0 | 25 | 1 |
| | 3b | | 0 | 0 | 0 | 8 | 0 | 4 | 0 |
| | 3f | | 0 | 10 | 0 | 39 | 0 | 35 | 0 |

[a]The selective pressure sites of HN and F protein of HPIV1–3 sequenced in this study were calculated by clade/lineage/sub-lineage. A *P*-value of less than 0.05 for SLAC, FEL, and MEME, while Bayes factor/posterior probability of greater than 0.95 for FUBAR (7, 27).

strains lost both O-glycosylation sites at positions 103 and 453 or only one O-glycosylation site at position 453.

Among the nine newly sequenced HPIV2 strains, nine N-glycosylated sites were conserved in most strains at amino acid positions 6, 272, 284, 316, 335, 341, 454, 501, and 517. Notably, HPIV2/CHN/BCH20170272/2017 was an exception, as it lacked an N-glycosylation site at position 316. O-glycosylated sites in the HN protein were predicted at aa positions 325, 326, and 442, whereas some strains presented additional predicted O-glycosylation sites at distinct positions. All HPIV2 strains harbored five potential N-glycosylation sites in the F protein at specific positions (65, 69, 77, 90, and 431). Furthermore, six strains exhibited O-glycosylation at position 156, whereas the remaining strains did not.

For the 101 newly sequenced HPIV3 strains, the HN protein of HPIV3 presented a conserved pattern of N-glycosylation, with three sites at amino acid positions 308, 485, and 523, whereas two strains identified in Ningxia Hui Autonomous Region presented an additional site at amino acid position 30. In addition, most of the HN protein sequences presented a conserved pattern of five putative O-glycosylation sites located at amino acid positions 126, 143, 352, 353, and 359. Notably, a single strain, HPIV3/CHN/BCH20190558/2019, displayed a distinct pattern, with two putative O-glycosylation sites found at amino acid positions 161 and 165. The F protein of HPIV3 presented four potential N-glycosylation sites (aa positions 238, 359, 446, and 508) in most strains, whereas three strains lacked one site at either aa position 446 or 508. The F protein of all HPIV3 strains possessed two O-glycosylated sites at aa positions 245 and 246, with the exception of one strain, which had an additional site at aa position 99.

Moreover, the HN protein of the single HPIV4b strain analyzed in this study presented five predicted N-glycosylation sites (aa positions 279, 339, 347, 502, and 530) and four predicted O-glycosylation sites (aa positions 340, 346, 350, and 354). The F protein of this HPIV4b strain contained three potential N-glycosylation sites (aa positions 66, 74, and 244) and one O-glycosylated site (aa position 439). All the predicted N-glycosylation and O-glycosylation sites are shown in Table S5.

## Entropy plots of the HPIV1–4 protein sequences

The information entropy of various positions was calculated by utilizing the aligned sequences, as depicted in Fig. 5. Our findings revealed that HPIV3 presented the greatest

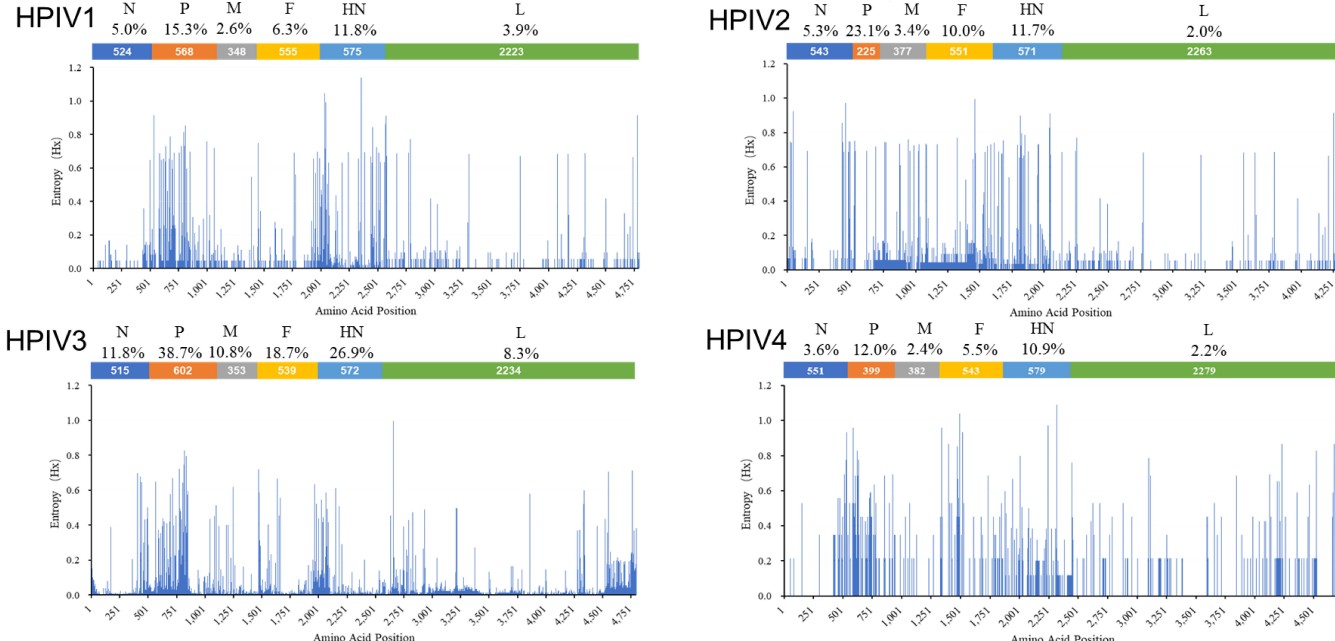

**FIG 5** Entropy plots of concatenated HPIV1 (A), HPIV2 (B), HPIV3 (C), and HPIV4b (D) protein sequences. For each amino acid position in the protein sequence, higher entropy values represent greater amino acid variation. Abbreviated protein sequence names are shown across the top of the plot in the order in which coding sequences are arranged in the genome. The percentage values show the percentage of positions in the protein that have a mutation present in more than one sequence.

variability across all proteins, whereas HPIV4 presented the least variability. Interestingly, our analysis revealed that the P protein presented greater diversity than the other proteins across all HPIV types. The two glycoproteins, HN and F, displayed relatively high variability in HPIV1–4, whereas the L proteins presented the least diversity.

## Amino acid substitutions in HN proteins of HPIV1–4

To investigate the amino acid substitutions in the HN proteins of the newly sequenced HPIVs, the HN genes of the four HPIV types were compared to those of their corresponding prototype strains. Most of the observed mutations presented clade-scale patterns, which were found in all or most of the strains within a clade, lineage, or sublineage (Table 2; Tables S6 to S9). The HN protein of HPIV1 is divided into seven regions, namely, the cytoplasmic tail, transmembrane region, stalk region, high structural homology region I, nonstructural assignment region, high structural homology region II, and carboxyl terminus (28), which exhibit 11, 11, 19, 9, 14, 4, and 11 amino acid substitutions, respectively. Among these regions, the transmembrane region and cytoplasmic tail presented the highest mutation rates of 31.4% and 44.0%, respectively, whereas the highly structural homology regions I and II were the most conserved, with substitution rates of only 6.0% and 8.1%, respectively. Similarly, the HN protein of HPIV3 comprises four regions, including the cytoplasmic tail, transmembrane region, stalk region, and head region (23), with the head region exhibiting the lowest mutation rate (6.7%) and the cytoplasmic tail showing the highest substitution rate (41.9%).

## Relationships among amino acid substitutions, negatively selected sites, and glycosylation sites in the HN protein of HPIV3

Coelingh et al. reported the presence of 13 murine monoclonal antibody-binding sites on the HN protein of HPIV3 (corresponding to amino acid residues in the HN protein of the prototype strain JN089924 HPIV3/USA/Washington/1957: 171K, 278S, 281A, 345N, 346E, 347N, 364N, 369S, 370P, 395K, 397W, 461N, and 500K) (29). These

**TABLE 2** Consensus amino acid substitutions for the clade, lineage, or sublineage in HN protein of HPIV1–4 sequenced in this study[a]

| Virus | Clade/lineage /sub-lineage | Amino acid mutations in HN |
|---|---|---|
| HPIV1 | B | N8I, V25A, H34Y/N/C, T45A, V46A, F49L, M76V, I82T, R131K, S151T, I245V, I335V, **D349N\***, N355K, R356S, T358P, R385H, L439I, N443K, K448N, E453K, Q461P, Y466F, R468K, V489F, N511S, E514K, V524G, A553T, I573V |
| | C | N8I, S20F, T42A, T45A, V46T, F49L, I70T, M76V, I82T, R131K, S151T, T187S, N332D, I335V, D349N, N355S, R356K, T358P, N443K, K448N, A450T, E453R, Q461P, Y466F, R468K, E514K, V524G, Q525K, A553T, L558F, I570V |
| | D | N8I, V22A, T28I, G31R, H34Y, T45A, V46A, F49L, I59V, M66I, M76V, I82T, R131K, S151T, I245V, I335V, D349N, N355D, R356S, T358P, N443K, K448D/N, E453K, Q461P, R468E, V489F, N511S, V524E, E527K, A553T, I573V |
| HPIV2 | G3 | D54N, I67V, F100L, N164H, I175S, S316N, K323E, K332T, K341N, Q345R, S351G, N360Y, V367I, H376Q, A416S, P479L, R497K, S513N, A514S |
| | G1b | D54N, I87V, F100L, T114A, V137A, K139E, N164H, T195A, A201S, A211G, S316N, P319T, K323E, K341N, E344K/T, A348I, A378E, R379E, D402G, A416S, D476N, P479L/I, T480M, Q482R, R497K, S513N, A514S, I570M, P571L |
| HPIV3 | C3a | **M21T\***, I28L, I40T, I53T, H62R, V69I, M82V, I87T, M118I, H295Y, I391V, D556N |
| | 3b | A13V, **M21T\***, G25S, I28L, I40T, I53T, H62R, V69I, M82V, I87T, M118I, H295Y, I391V, D556N |
| | 3f | **M21T\***, I28L, I40T, I53T, H62R, V69I, M82V, I87T, M118I, H295Y, I391V, D556N |
| HPIV4 | 4b | D3E, V35I, N55S, H56D, I57V, N58D, R82A, S96R, I100T, A116V, S126G, R128K, V133A, S156P, A198E, V201A, N204K, G233R, P285A, D286H, H306W, R333G, S348R, K354T, R367G, Y396H, S413F, P434S, S435N, T442I, Q443E, I444T/V, S453P, E506A, V572R, N579T |

[a]Amino acid positions correspond to the prototype strain (AF457102.1_HPIV1/USA Washington/1964, AF533012_HPIV2/USA/GREER/1955, JN089924.1_HPIV3_USA/Washington/1957 and AB543337.1_HPIV4b/Japan Tokyo/68–333/1968). Negative selection sites were highlighted with bold fonts and labeled by "*".

monoclonal antibody-binding sites are likely associated with the formation of B-cell epitopes on the HN protein (30). To investigate the relationships among amino acid substitutions, negatively selected sites, glycosylation sites, and B-cell epitopes in the HN protein of HPIV3, a total of 101 newly sequenced HPIV3 HN sequences were analyzed. Our findings indicated the presence of two amino acid substitutions, N461D and K500R, which correspond to reported neutralization-related sites detected in HPIV3/CHN/GY17027/2017 and HPIV3/CHN/BCH20151067/2015, respectively. Both of these substitutions belong to lineages C3f and C3b, respectively (Fig. 6). However, we did not identify any glycosylated site or selected pressure site that corresponds to any reported mAb neutralizing site in the HN protein of our HPIV3 strains.

## DISCUSSION

HPIVs are prevalent worldwide and pose a significant threat to vulnerable populations of all ages, especially young children and elderly individuals (3, 11, 20). The lack of available vaccines or antiviral treatments for HPIVs is a significant challenge in clinical management (11). The limited genomic sequences of HPIVs in GenBank, especially whole-genome sequences, and the inadequate surveillance system further hinder the development of effective vaccines and antiviral drugs, as well as the study of the pathogenic mechanism underlying HPIVs (3, 31). To address this issue, this study utilized the Illumina MiSeq platform to sequence HPIV-positive samples collected from hospitalized children diagnosed with ALRTI in China between 2015 and 2021. A total of 156 complete genome sequences, including 45 HPIV1, 9 HPIV2, 101 HPIV3, and 1 HPIV4b, presented high nucleotide and amino acid identities within the same type.

The HN gene has been widely recognized as the primary target for conducting phylogenetic analyses of HPIVs because of its relatively high levels of sequence and antigenic variation. In this study, the phylogenetic tree based on the HN CDS of HPIV1 revealed four distinct clades, A, B, C, and D, with clades B and D showing greater diversity, which was similar to the classification results of Shao et al. (3) and Li et al. (32). All the newly sequenced HPIV1 strains were categorized into three distinct clades, namely, B, C, and D. Notably, clade B was found to be predominant in China and demonstrated a high degree of genetic similarity with strains reported in various countries, including Malaysia, Thailand, France, Germany, Croatia, the Netherlands, and the United States. Interestingly, the eight strains identified in Beijing and Zhejiang Provinces grouped with a single HPIV1

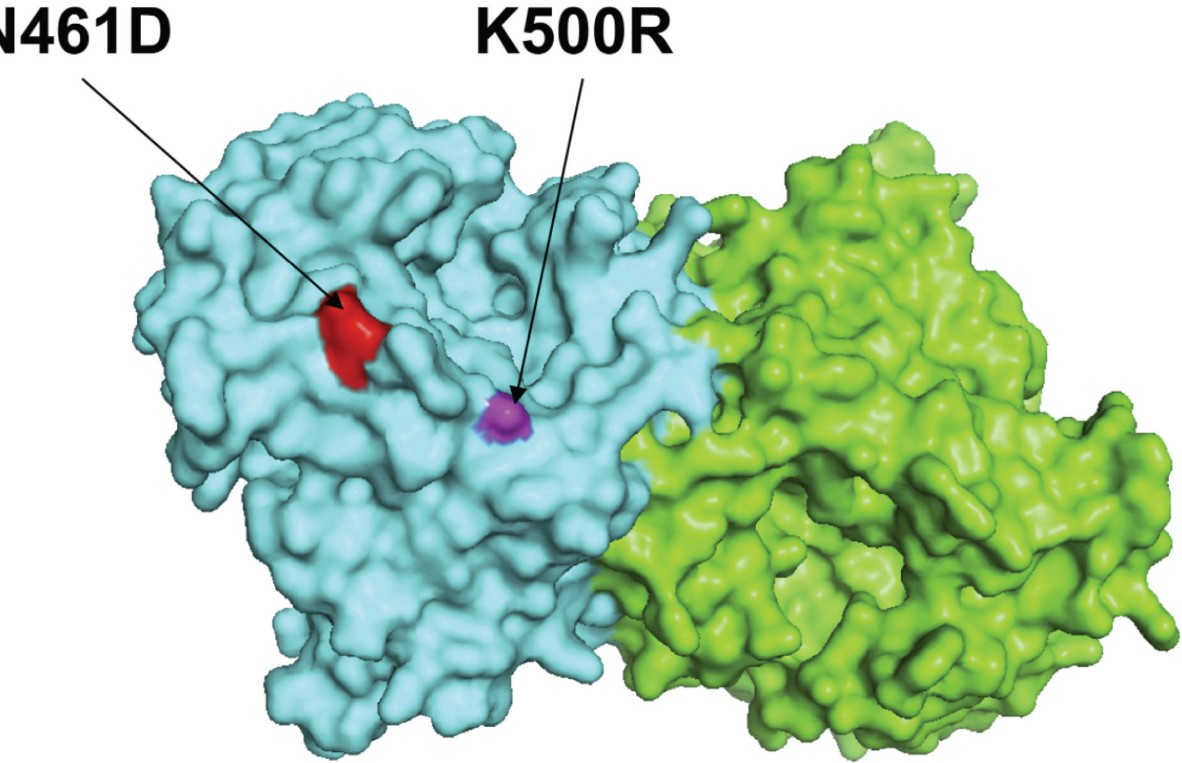

**FIG 6** The two amino acid substitution sites corresponding to reported neutralizing mAb sites in the HN protein of newly sequenced HPIV3 strains were mapped on the structure model of prototype strain Washington 1957. Template for homology modeling was using the crystal structure of 4xjq. Chains of the dimer structure model were colored in green (chain A) and cyan (chain B). The N461D and K500R amino acid substitution sites were labeled within red and white color, respectively.

strain isolated from Jilin Province, indicating the emergence of a novel clade, designated D, which appeared to be geographically restricted to China.

With respect to HPIV2, Šantak et al. revealed that lineage G1a strains emerged as the dominant cluster in Croatia between 2011 and 2017, gradually replacing clade G3 strains (33, 34). This genotype shift may be attributed to the limited cross-protection ability of neutralizing antibodies against the two clusters (33). Previous studies have reported limited sequences of HPIV2 strains. Currently, the predominant global HPIV2 strains belong to lineages G3 and G1. Lineage G3 emerged in 1998 with the V98 strain (AF533011_HPIV2/USA/V98/1998) in the United States and has since continued to circulate in countries such as the United States, China, Vietnam, Malaysia, Croatia, and Italy, maintaining a high level of genetic diversity (within-group p-distance of $0.01294 \pm 0.00132$). Within clade G1, lineage G1a began with the earliest strain V94 (AF533010_HPIV2/USA/V94/1994) from the United States and has gradually become a major circulating strain in the United States, China, France, Croatia, Japan, and Germany, also exhibiting significant genetic diversity (within-group p-distance of $0.01423 \pm 0.00127$). Currently, lineages G1a and clade G3 are the predominant circulating genotypes of HPIV2 in China and globally.

To date, most phylogenetic analyses of HPIV3 have adopted the classification system proposed by Mao et al. (23) and subsequently refined by Almajhdi (35) et al. and Aso et al. (36). Based on this system, all HPIV3 strains can be divided into three distinct clades (A, B, and C) using the HN gene as the primary target. Clades A and B consist of strains mainly found in the last century, which either became less prevalent or appeared to have died off after 2000 (7). Conversely, the strains belonging to clade C are highly diverse and are currently prevalent globally. In the absence of any HPIV3 strains sequenced in China before 2000, all HPIV3 sequences available for Chinese strains belong to clade C. Within

this clade, HPIV3 strains clustered into lineages C3a, C3b, C3c, C3e, and C3f, with only one exception being found in lineage C1c. In this study, all 101 HPIV3 strains analyzed were found to belong to lineages C3a, C3b, and C3f, which displayed close similarity with isolates circulating in Asia, America, and Europe. These findings demonstrated that several HPIV3 lineages cocirculated in China, with lineages C3a, C3b, and C3f being the most prevalent, which is consistent with previous studies (3, 37). Furthermore, based on the HN phylogenetic tree, we identified a novel lineage formed by strains from Pakistan and Russia in 2023 within clade C. These strains exhibit substantial genetic distance from other lineages within clade C. Consistent results were observed in both the Bayesian and ML phylogenetic trees for HN. Therefore, we designated this new evolutionary branch within clade C as lineage C6. Continued monitoring and research are needed to determine whether this new lineage will continue to evolve and become prevalent in the region or globally or whether it will diminish rapidly, similar to the strains in lineage C4.

Overall, the evolutionary rates of HPIV HN genes are relatively low, hovering around the level of 1.0E−04 substitutions/site/year, with HPIV1 and HPIV3 exhibiting the highest and lowest evolutionary rates, respectively. This finding contrasts with those by Linster et al., in which the evolutionary rate of HPIV4 was highest, followed by those of HPIV1, HPIV2, and HPIV3. Such discrepancies may arise from strain selection and the different calculation methods employed. To date, no studies have simultaneously compared the WGS evolutionary rates of HPIV1–4. This study revealed that HPIV4 has the highest WGS evolutionary rate among all HPIV types, providing fundamental data for evolutionary studies of HPIV1–4. Notably, the evolutionary rates of both HN and WGS for HPIV3 are the lowest among the four types, at 5.85E−04 substitutions/site/year, which aligns closely with previous studies on the evolutionary rates of HPIV3 whole genomes ranging from 3.59E−04 to 4.2E−04. HPIV4 strains have been sporadically identified in various regions (8, 20). Nevertheless, few publications have documented the circulation of HPIV4 strains in mainland China (3, 12, 24, 37, 38). One of these studies reported multiple lineages of HPIV4 in Henan Province between 2017 and 2018 (24), whereas another detected two HPIV4a strains in Beijing in 2014 (3). In our investigation, we obtained the first complete genome sequence of an HPIV4b strain in mainland China, which was classified into lineage C2 of HPIV4b. This strain is closely related to two Henan strains and to strains from Japan, the United States, and Germany.

In previous studies, alternative genes such as the F gene were utilized to perform phylogenetic analyses of HPIVs, but the potential of employing whole-genome sequences to elucidate the molecular and evolutionary characteristics of these viruses has not been fully explored (3, 7, 32, 34, 39). Therefore, one of the objectives of our study was to assess the efficacy of utilizing whole-genome sequences to construct phylogenetic topologies similar to those generated using the HN gene. Our findings demonstrated that whole-genome sequences would be a practical and effective tool for conducting phylogenetic analyses of HPIVs, thereby expanding the available options for researchers in this field.

Recombination and mutation are vital mechanisms for the evolution of RNA viruses (1). Previous studies reported recombination events for HPIV3 (3, 7). No recombination events were identified in HPIV2 or HPIV4 strains before and in our study, which was likely due to the limited whole-genome sequences of these viruses deposited in the GenBank database. In our study, only one HPIV1 strain, which belongs to clade B, was inferred to have undergone recombination events across part of the F gene.

Paramyxoviruses share a genome structure and replication strategy. Their genomes are all organized to encode at least six common structural proteins (3′-N-P-C-M-F-HN-L-5′) (1). The N protein can bind to viral RNA, which is a template that allows the P and L proteins to be transcribed and eventually replicate the genome of HPIV (7, 40). The M protein plays a vital role in coordinating assembly by interacting with both the cytoplasmic tails of viral glycoproteins and the viral ribonucleoprotein complex and budding new virions (1, 41, 42). The HN and F proteins, both of which are surface glycoproteins, cooperate in a highly specific way to mediate fusion upon receptor-binding during

virus entry (43). The multifunctional L protein is the major polymerase subunit and is required for RNA synthesis, mRNA capping, methylation, and polyadenylation. Therefore, we calculated the entropy values of the six genes of HPIV1—4. Interestingly, the results revealed that all six viral proteins of HPIV3 presented more variable sites than did the corresponding proteins in the other three HPIVs. The increased percentage of variable sites may at least partially be due to HPIV3 having the most available sequences in GenBank, allowing for a greater amount of population variation to be found. In addition, the L protein clearly presented the fewest mutations in HPIVs, with the exception of HPIV1, which had similar results to those of a previous study (7), and the P protein presented the most variations among all types of HPIV. This protein is not well conserved between HPIV1-4, which suggests that the P gene may also be useful as a molecular marker for evolutionary analyses and epidemiology research. The entropy analysis also highlights more conserved regions of the genome that may be useful for the development of robust diagnostics and potential vaccines.

Glycosylation is an essential posttranslational modification that plays a pivotal role in influencing protein folding, antigenicity, and biological activity (3, 44). The F and HN proteins of HPIVs are widely recognized as crucial players in viral infection (9, 36, 45). Our findings revealed a high degree of conservation in the glycosylation patterns of these proteins of each HPIV type, which was consistent with previous studies (3, 33, 46). Notably, an extra glycosylation site (aa position 511) at the carboxy terminus of the HN protein was present in four newly sequenced clade C HPIV1 strains, which shares characteristics similar to those of clade C strains from other regions. Whether the glycosylated site at amino acid position 511 on the HN protein of HPIV1 has potential implications for both the neutralizing activity of the antibody against the virus and viral pathogenicity needs further study in the future.

Owing to their inability to correct viral RNA-dependent RNA polymerases, RNA viruses present a high genetic mutation rate (3). Any mutation in a viral protein, especially the HN protein, may affect its function and activity. Hence, we analyzed the relationships among amino acid substitutions, negatively selected sites, and glycosylation sites in the HN protein of HPIV3. We found that two HPIV3 strains possessed two amino acid substitutions (N461D and K500R) in the HN protein of HPIV3 corresponding to two reported neutralization-related sites (29), which might be one reason for the reinfection of HPIV3 in children. Previous studies have indicated that the majority of B-cell epitopes within the HN protein of HPIV3 are likely associated with conformational epitopes (30). Aso et al. identified conformational epitopes in the HN protein of HPIV3 that differed from previously reported linear epitope motifs (36). Therefore, further research is needed to understand the impact of these mutations on viral evasion of host immunity.

However, the present study has certain limitations. First, the newly sequenced HPIV strains from the six regions in mainland China may not be sufficient to capture the full spectrum of evolutionary patterns of HPIV1–4 in other regions of China. Second, the differences in HPIV1–4 prevalence observed in this study are likely due to the selection of patients with severe respiratory illness. HPIV2 and HPIV4, known for causing milder infections, are expected to be underrepresented among those with ALRTIs included in the study. Therefore, this study may not fully capture the complete evolutionary pattern of HPIV1–4 in China. Third, given the limited availability of HPIV2 and HPIV4 genomic sequences in China, an exhaustive understanding of the evolutionary traits of the two HPIV types within the Chinese population is impeded.

In summary, our study provides a comprehensive genetic analysis of HPIV1–4 circulating among children in China between 2015 and 2021 and identifies a novel clade of HPIV1, namely, clade D. These findings represent a significant advancement in the understanding of the evolutionary patterns of HPIV1–4 in China and provide important insights for further research on molecular diagnosis, vaccine development, and surveillance of HPIVs. The establishment of an efficient surveillance system for HPIVs and the conduct of additional research on the genetic variation and evolutionary trends

of the virus based on whole-genome sequences will be crucial for the prevention and control of HPIV-related diseases in China.

## ACKNOWLEDGMENTS

We would like to thank all participating members of hospitals for their assistance and collaboration in the samples and clinical data collection. We are grateful to Professor Naiying Mao 's team in Chinese Center for Disease Control and Prevention (China CDC) for their generous assistance with Bayesian phylogenetic analysis of the sequences.

This work was supported by The National Science and Technology Major Project (2017ZX10103004-004, 2017ZX10104001-005-010), Chinese Academy of Medical Sciences Innovation Fund for Medical Sciences (2019-I2M-5-026), The Capital Health Development and Research of Special (2021-1G-3012), and Funding for Reform and Development of Beijing Municipal Health Commission. These sponsors had no role in the study design, data analysis, manuscript preparation, or decision to publish.

Z.X., X.C., and Y.Z. conceived and designed the study; Y.Z., X.C., J.A., R.W., and Y.D. performed the experiments; L.L., L.C., G.L., R.J., B.X., and Y.S. provided clinical specimens and analyzed the data; Y.Z. drafted the manuscript. Z.X. and X.C. revised the manuscript. All authors read and approved the final manuscript.

The authors have indicated they have no financial relationships relevant to this article to disclose.

## AUTHOR AFFILIATIONS

[1]Laboratory of Infection and Virology, Beijing Pediatric Research Institute, Beijing Key Laboratory of Pediatric Respiratory Infection Diseases, Beijing Children's Hospital, Capital Medical University, Key Laboratory of Major Diseases in Children, Ministry of Education, National Clinical Research Center for Respiratory Diseases, National Center for Children's Health, Beijing, China
[2]Research Unit of Critical infection in Children, Chinese Academy of Medical Sciences, Beijing, China
[3]The Division of General Pediatrics, Yinchuan Women and Children Healthcare Hospital, Yinchuan, China
[4]Department of Pediatric of Pulmonology, The 2nd Affiliated Hospital and Yuying Children's Hospital, Wenzhou Medical University, Wenzhou, China
[5]The Respiratory Department, Guangzhou Women and Children's Medical Center, Guangzhou, China
[6]The Respiratory Department, Guizhou Maternal and Child Health Care Hospital, Guiyang Children's Hospital, Guiyang, China
[7]Department of Respiratory Diseases I, Beijing Children's Hospital, Capital Medical University, National Clinical Research Center for Respiratory Diseases, National Center for Children's Health, Beijing, China
[8]The Division of Pediatric Respiratory Medicine, Shengjing Hospital of China Medical University, Shenyang, China

## AUTHOR ORCIDs

Yun Zhu ⓘ http://orcid.org/0000-0003-3531-3544
Ran Wang ⓘ https://orcid.org/0000-0002-0243-8538
Xiangpeng Chen ⓘ http://orcid.org/0000-0001-6955-6515
Zhengde Xie ⓘ http://orcid.org/0000-0001-7634-9338

## FUNDING

| Funder | Grant(s) | Author(s) |
| --- | --- | --- |
| National Science and Technology Major Project (国家科技重大专项) | 2017ZX10103004-004 | Zhengde Xie |

| Funder | Grant(s) | Author(s) |
|---|---|---|
| National Science and Technology Major Project (国家科技重大专项) | 2017ZX10104001-005-010 | Xiangpeng Chen |
| The Capital Health Devolepment and Research of Special | 2021-1G-3012 | Yun Zhu |
| Funding for Reform and Development of Beijing Municipal Health Commission | | |

## AUTHOR CONTRIBUTIONS

Yun Sun, Investigation, Methodology, Project administration, Writing – review and editing | Changchong Li, Investigation, Methodology, Project administration, Writing – review and editing | Gen Lu, Investigation, Methodology, Project administration, Writing – review and editing | Rong Jin, Investigation, Methodology, Project administration, Resources, Writing – review and editing | Baoping Xu, Investigation, Methodology, Project administration, Writing – review and editing | Yunxiao Shang, Investigation, Methodology, Project administration, Resources, Writing – review and editing | Junhong Ai, Data curation, Investigation | Ran Wang, Data curation, Investigation, Writing – review and editing | Yali Duan, Data curation, Investigation | Xiangpeng Chen, Investigation, Writing – review and editing | Zhengde Xie, Funding acquisition, Writing – review and editing.

## ETHICS APPROVAL

The study was granted ethical approval by the Ethical Review Committee of Beijing Children's Hospital (Ethics number: 2017k-15).

## ADDITIONAL FILES

The following material is available online.

### Supplemental Material

**Figure S1 (Spectrum03432-23-s0001.pdf).** ML trees of HN of HPIV1-4.
**Figure S2 (Spectrum03432-23-s0002.pdf).** ML trees of WGS of HPIV1-4.
**Figure S3 (Spectrum03432-23-s0003.pdf).** Genetic distance with group and between groups of HPIV1-4.
**Tables S3-S5 (Spectrum03432-23-s0006.docx).** NSS, glycosylation site.
**Table S1 (Spectrum03432-23-s0004.xlsx).** Stains in this study.
**Table S2 (Spectrum03432-23-s0005.xlsx).** Reference strains.
**Table S6 (Spectrum03432-23-s0007.xlsx).** Amino acid mutations of HPIV1.
**Table S7 (Spectrum03432-23-s0008.xlsx).** Amino acid mutations of HPIV2.
**Table S8 (Spectrum03432-23-s0009.xlsx).** Amino acid mutations of HPIV3.
**Table S9 (Spectrum03432-23-s0010.xlsx).** Amino acid mutations of HPIV4.

### Open Peer Review

**PEER REVIEW HISTORY (review-history.pdf).** An accounting of the reviewer comments and feedback.

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
