## [Reviewer comments · Microbiology Spectrum]

Microbiology Spectrum

Genetic characteristics of human parainfluenza viruses 1-4 associated with acute lower respiratory tract infection in Chinese children, during 2015-2021

Yun Zhu, Yun Sun, Changchong Li, Gen Lu, Baoping Xu, Rong Jin, Yunxiao Shang, Junhong Ai, Ran Wang, Yali Duan, Xiangpeng Chen, and Zhengde Xie

Corresponding Author(s): Zhengde Xie, Beijing Pediatric Research Institute, Beijing Children's Hospital, Capital Medical University

Review Timeline:

Submission Date:	September 21, 2023
Editorial Decision:	October 23, 2023
Revision Received:	May 12, 2024
Editorial Decision:	June 8, 2024
Revision Received:	August 6, 2024
Accepted:	August 9, 2024

Editor: Biao He

Reviewer(s): Disclosure of reviewer identity is with reference to reviewer comments included in decision letter(s). The following individuals involved in review of your submission have agreed to reveal their identity: Tanushree Dangi (Reviewer #2)

Transaction Report:

DOI: <https://doi.org/10.1128/spectrum.03432-23>

Re: Spectrum03432-23 (Genetic characteristics of human parainfluenza viruses 1-4 associated with acute lower respiratory tract infection in Chinese children, during 2015-2021)

Dear Prof. Zhengde Xie:

Thank you for the privilege of reviewing your work. Below you will find my comments, instructions from the Spectrum editorial office, and the reviewer comments.

Please provide Abstract and Significance in the manuscript, the methods and results sections should also be described adequately.

Revision Guidelines

Sincerely,
Biao He
Editor
Microbiology Spectrum

Reviewer #1 (Comments for the Author):

Journal: Microbiology Spectrum

Type of manuscript: Research

Title: Genetic characteristics of human parainfluenza viruses 1-4 associated with acute lower respiratory tract infection in Chinese children, during 2015-2021

Summary/General Reviewer Comments: The manuscript describes the molecular analysis of 156 complete genome sequences of HPIVs obtained from children hospitalized with ALRTI from 2015 to 2021 in China. A comprehensive analysis of the F and HN genes and their predicted glycoproteins is also presented, as these are likely to underpin future HPIV vaccines and antivirals. The data are novel and valuable and the analysis is comprehensive. Though generally well written, the manuscript could benefit from some grammatical correction and text shortening (examples proved). Note that the Abstract is missing from the manuscript.

Specific Reviewer Comments:

- 1) Introduction, Line 53. Change (italicize) Paramyxovirus to Paramyxoviridae.
- 2) Introduction, Line 64-66. Change to, "Of these, the HN and F surface glycoproteins mediate host cell receptor binding and viral entry and are principle targets of the specific host immune response."
- 3) Introduction, Line 67. Spell out "ALRTI" when acronym first used.
- 4) Introduction, Line 67-83. This section can be shortened and reorganized to (for example):

HPIVs are a significant cause of acute lower respiratory tract infection (ALRTI) among children under the age of five (Wang et al., 2021; Zhu et al., 2021), contributing globally to 4-14% of ALRTI hospital admissions and 4% of childhood ALRTI related deaths (Wang et al., 2021). HPIV infections have also been associated with upper and lower respiratory tract illnesses, including the common cold, croup, tracheobronchitis, bronchiolitis, and pneumonia, in both children and adults (Diaz-Chiguer et al., 2019; Li et al., 2019). Although these infections are typically mild in healthy individuals, they may lead to severe respiratory diseases in infants, the elderly and immunocompromised individuals (Greninger et al., 2021; Maeda et al., 2017). Individuals are susceptible to recurrent HPIV infections throughout their life due to incomplete protective immunity to these viruses (Bernstein et al., 2012). Serological studies have shown that up to 80% of children are infected with HPIV3 by the age of four (Linster et al., 2018; Mackenzie et al., 2019; Oh et al., 2021).

- 5) Introduction, Line 73. What is "[7]"?
- 6) Materials & Methods, Line 96. Change "During 2015 and 2021," to "From 2015 to 2021,".
- 7) Materials & Methods, Line 116-117. Change to "Sequence data underwent quality control analysis using FastQC software, while genome assembly was performed using the optimized ..."
- 8) Materials & Methods, Line 122. Spell out CDS here first, not on line 130.
- 9) Materials & Methods, Line 125-126. "low quality sequences were excluded."
- 10) Materials & Methods, Line 147-149. Change to, "In order to identify potential positive (PSS) and negative (NSS) selection sites in the HPIV1-3 HN and F proteins ..."
- 11) Results, Line 183. Clarify if identity scores were obtained using all sequences (obtained in this study and previously published) or only those obtained in this study.
- 12) Results, Lines 235-236. Don't repeat Methods here. Change to, "No HPIV2, HPIV3 or HPIV4b strains sequenced in this study showed evidence of recombination."
- 13) Results, Line 240. Remove "obtained in this study".
- 14) Results, Lines 250-252. Don't repeat Methods here. Change to, "Selective pressure analysis found no PSS in the HPIV1-3 HN and F genes, but multiple NSS sites were inferred by at least two of the four algorithms (SLAC, FLE, MEM and Fubar) (Table 1; Supplementary Tables 6 and 7)."
- 15) Discussion, Line 431. Change to "Paramyxoviruses share genome structure and replication strategy."
- 16) Discussion, Line 464. Change "carbon terminal" to "carboxy-terminal".
- 17) Discussion, Line 484. The most likely explanation for the differences in HPIV1-HPIV4 prevalence in this study is the selection of cases with more severe respiratory illness. HPIV2 and HPIV4 have generally been associated with milder infections, and would be expected to be less represented among those enrolled with ALRTI.
- 18) Discussion, Lines 375-377 & 385-387. Remove sentences beginning with "Further ...". Not essential.

Reviewer #2 (Comments for the Author):

Manuscript ID: Spectrum03432-23

Title: Genetic characteristics of human parainfluenza 1 viruses 1-4 associated with acute lower respiratory tract infection in Chinese children, during 2015-2021

This study identified the four types of human parainfluenza viruses (HPIV) in a cohort of children affected with acute lower respiratory tract infection and characterized these strains based on haemagglutinin-neuraminidase (HN) and whole genome

sequences. The author has reported the circulation of multiple strains of HPIV1-4 in the population with the emergence of novel B4 substrain of HPIV1, highlighting the significance of continuous surveillance of HPIV. The strength of this article is that it uses a significant number of clinical samples to investigate the molecular diversity and evolutionary pattern of HPIV by performing whole genome sequencing, which is difficult to manage, especially isolated from children.

However, the methods are poorly written and not described adequately. Similarly, the few sections of the results (see details in comments) are unclear and elaborated superficially. I would suggest updating this article thoroughly, especially the methods and results.

Please find the detailed concerns listed below.

Abstract is missing in the actual text.

Introduction: The author has cited the below statement with reference 7 . I don't see reference number 7 in the list. Please include the reference in the right format in the text like other references instead of showing it in numbers.

Notably, the burden of HPIVs in children is comparable to that of RSV infection, although the incidence of ALRTI among different age groups varies widely among populations [7].

There is no clarity in the methodology section. How was the library constructed and sequencing performed? Please briefly describe the sequencing method.

How many and what reference strains were selected for each type of HPIV for mutational and phylogenetic analysis? Please include the details about reference strains used for comparative nucleotide analysis and for /phylogeny.

How the complete genome of HPIV was assembled. Please describe briefly in the methods.

What parameters were considered for interpreting the recombinant events in boot scanning and Simplot analysis? Please include this information in the methods.

It is mentioned in the results that All the HPIV1 strains could be categorized into three distinct clades (clade A-C), with the subdivision of clade B into four sub-clades. In what reference, study strains were categorized into clades or subclades? No details are provided in the text.

It is unclear from the text whether sequence similarity analysis was performed on the full-length genome or only for the HN coding sequence.

The common amino acid substitution on the HN protein is shown only for HPIV1 & HPIV3 but not for HPIV2 & HPIV4. Why?

The author reported that only the HPIV 1 strain showed a recombination event in the F gene, confirmed by Simplot/bootscan analysis in figure 3. However, it is not very clear in figure 3, how the recombination event was determined. Please elaborate figure in the result section and also highlight the recombination region in the figure.

What are the PSS and NSSs in the table/ and what is the significance of considering these parameters? Please explain these parameters in methods/results. Please expand the abbreviated form in the legend section of the table.

I wonder why this study has not estimated the evolutionary rate of all HPIV types based on HF and the complete genome, which is an important parameter if someone is investigating strain diversity in context to time. Please include this analysis.

Figure 6 shows two amino acid mutations, N461D and K500R determined in HPIV3 strains, which corresponds to the neutralizing mAb sites in the HN protein. How these mutations were mapped on the prototype strain is not mentioned anywhere in the text. Please include this information in the text.

What are the A, B1-B4, and C represent in the supp. table 2? Similarly, what do the G1a-G1c, G2, G3, and G4a-G4b represent in supp. table 3 and other supp tables? Are these reference strains or the study strains? Please include the statements about whether the evolutionary distances were calculated among study strains or among the prototype strains.

Manuscript ID: Spectrum03432-23

Title: Genetic characteristics of human parainfluenza 1 viruses 1-4 associated with acute lower respiratory tract infection in Chinese children, during 2015-2021

This study identified the four types of human parainfluenza viruses (HPIV) in a cohort of children affected with acute lower respiratory tract infection and characterized these strains based on haemagglutinin-neuraminidase (HN) and whole genome sequences. The author has reported the circulation of multiple strains of HPIV1-4 in the population with the emergence of novel B4 substrain of HPIV1, highlighting the significance of continuous surveillance of HPIV. The strength of this article is that it uses a significant number of clinical samples to investigate the molecular diversity and evolutionary pattern of HPIV by performing whole genome sequencing, which is difficult to manage, especially isolated from children.

However, the methods are poorly written and not described adequately. Similarly, the few sections of the results (see details in comments) are unclear and elaborated superficially. I would suggest updating this article thoroughly, especially the methods and results.

Please find the detailed concerns listed below.

Abstract is missing in the actual text.

Introduction: The author has cited the below statement with reference 7 . I don't see reference number 7 in the list. Please include the reference in the right format in the text like other references instead of showing it in numbers.

Notably, the burden of HPIVs in children is comparable to that of RSV infection, although the incidence of ALRTI among different age groups varies widely among populations [7].

There is no clarity in the methodology section. How was the library constructed and sequencing performed? Please briefly describe the sequencing method.

How many and what reference strains were selected for each type of HPIV for mutational and phylogenetic analysis? Please include the details about reference strains used for comparative nucleotide analysis and for /phylogeny.

How the complete genome of HPIV was assembled. Please describe briefly in the methods.

What parameters were considered for interpreting the recombinant events in boot scanning and Simplot analysis? Please include this information in the methods.

It is mentioned in the results that All the HPIV1 strains could be categorized into three distinct clades (clade A-C), with the subdivision of clade B into four sub-clades. In what reference, study strains were categorized into clades or subclades? No details are provided in the text.

It is unclear from the text whether sequence similarity analysis was performed on the full-length genome or only for the HN coding sequence.

The common amino acid substitution on the HN protein is shown only for HPIV1 & HPIV3 but not for HPIV2 & HPIV4. Why?

The author reported that only the HPIV 1 strain showed a recombination event in the F gene, confirmed by Simplot/bootscan analysis in figure 3. However, it is not very clear in figure 3, how the recombination event was determined. Please elaborate figure in the result section and also highlight the recombination region in the figure.

What are the PSS and NSSs in the table/ and what is the significance of considering these parameters? Please explain these parameters in methods/results. Please expand the abbreviated form in the legend section of the table.

I wonder why this study has not estimated the evolutionary rate of all HPIV types based on HF and the complete genome, which is an important parameter if someone is investigating strain diversity in context to time. Please include this analysis.

Figure 6 shows two amino acid mutations, N461D and K500R determined in HPIV3 strains, which corresponds to the neutralizing mAb sites in the HN protein. How these mutations were mapped on the prototype strain is not mentioned anywhere in the text. Please include this information in the text.

What are the A, B1-B4, and C represent in the supp. table 2? Similarly, what do the G1a-G1c, G2, G3, and G4a-G4b represent in supp. table 3 and other supp tables? Are

these reference strains or the study strains? Please include the statements about whether the evolutionary distances were calculated among study strains or among the prototype strains.

Response to Reviewers

Dear editor,

Thank you for your decision letter concerning our manuscript (ID Spectrum03432-23) entitled “Genetic characteristics of human parainfluenza viruses 1-4 associated with acute lower respiratory tract infection in Chinese children, during 2015-2021”, and your time regarding for our revision. I also appreciate all the critical comments from you and reviewers. We have carefully considered the comments and revised the manuscript accordingly. With these improvements, we hope that the current version can meet the Journal’s standards for publication. The following is a point-by-point response to all those comments and a list of changes we have made to the manuscript.

Sincerely,

Zhengde Xie, M.D. (xiezhengde@bch.com.cn)

Reviewer #1

Q1. Introduction, Line 53. Change (italicize) *Paramyxovirus* to *Paramyxoviridae*.

Answer: Thank you so much. We have corrected the word. (Page 4, line 101)

Q2. Introduction, Line 64-66. Change to, "Of these, the HN and F surface glycoproteins mediate host cell receptor binding and viral entry and are principle targets of the specific host immune response."

Answer: Thank you for your suggestion. We have revised the sentence. (Page 4, line 109-111)

Q3. Introduction, Line 67. Spell out "ALRTI" when acronym first used.

Answer: Thanks. Revised edition: Page 4, line 113.

Q4. Introduction, Line 67-83. This section can be shortened and reorganized to (for example): HPIVs are a significant cause of acute lower respiratory tract infection (ALRTI) among children under the age of five (Wang et al., 2021; Zhu, et al., 2021), contributing globally to 4-14% of ALRTI hospital admissions and 4% of childhood ALRTI related deaths (Wang et al., 2021). HPIV infections have also been associated with upper and lower respiratory tract illnesses, including the common cold, croup, tracheobronchitis, bronchiolitis, and pneumonia, in both children and adults (Diaz-Chiguer et al., 2019; Li et al., 2019). Although these infections are typically mild in healthy individuals, they may lead to severe respiratory diseases in infants, the elderly and immunocompromised individuals (Greninger et al., 2021; Maeda et al., 2017). Individuals are susceptible to recurrent HPIV infections throughout their life due to incomplete protective immunity to these viruses (Bernstein et al., 2012). Serological studies have shown that up to 80% of children are infected with HPIV3 by the age of four (Linster et al., 2018; Mackenzie et al., 2019; Oh et al., 2021).

Answer: Thanks for the nice revision suggestion. Revised edition: Page 4-5, lines 113-123.

Q5. Introduction, Line 73. What is "[7]"?

Answer: Thanks. We have rewritten this paragraph and cited the right reference in the revised edition. Page 4-5, lines 113-123.

Q6. Materials & Methods, Line 96. Change "During 2015 and 2021," to "From 2015 to 2021,".

Answer: Thanks. Revised edition: page 5, line 130.

Q7. Materials & Methods, Line 116-117. Change to "Sequence data underwent quality control analysis using FastQC software, while genome assembly was performed using the optimized ..."

Answer: Thanks for your suggestion, we have rewritten the sequencing and splicing methods to make the methods more detailed. Page 6, line 152-164 .

Q8. Materials & Methods, Line 122. Spell out CDS here first, not on line 130.

Answer: Thanks. Revised edition: page 6, line 175.

Q9. Materials & Methods, Line 125-126. "low quality sequences were excluded."

Answer: Thank you for your suggestion. We have rewritten this sentence, page 6-7, line 166-179.

Q10. Materials & Methods, Line 147-149. Change to, "In order to identify potential positive (PSS) and negative (NSS) selection sites in the HPIV1-3 HN and F proteins ..."

Answer: Thanks. We have revised this sentence in page 8, line 234-235.

Q11. Results, Line 183. Clarify if identity scores were obtained using all sequences (obtained in this study and previously published) or only those obtained in this study.

Answer: Thank you for your suggestions. We have analyzed the nucleotide identity of the whole genome sequences obtained from HPIV1-4 in this study, as well as the nucleotide homology between these strains and their prototype strains. In revised manuscript, we have clarified. Page 7, line 182-184.

Q12. Results, Lines 235-236. Don't repeat Methods here. Change to, "No HPIV2, HPIV3 or HPIV4b strains sequenced in this study showed evidence of recombination."

Answer: Thank you for your suggestion. We have removed redundant method descriptions in the revised manuscript.

Q13. Results, Line 240. Remove "obtained in this study".

Answer: Thank you, we have made a revision in manuscript.

Q14. Results, Lines 250-252. Don't repeat Methods here. Change to, "Selective pressure analysis found no PSS in the HPIV1-3 HN and F genes, but multiple NSS sites were inferred by at least two of the four algorithms (SLAC, FLE, MEM and Fubar) (Table 1; SupplementaryTables 6 and 7)."

Answer: Thank you for your suggestion. We have made revisions based on your advice. Page 13, line 368-382.

Q15. Discussion, Line 431. Change to "Paramyxoviruses share genome structure and replication strategy."

Answer: Thanks. We have revised this sentence in page 17, line 571.

Q16. Discussion, Line 464. Change "carbon terminal" to "carboxy-terminal".

Answer: Thanks. We have revised this word in page 21, line 599.

Q17. Discussion, Line 484. The most likely explanation for the differences in HPIV1-HPIV4 prevalence in this study is the selection of cases with more severe respiratory illness. HPIV2 and HPIV4 have generally been associated with milder infections, and would be expected to be less represented among those enrolled with ALRTI.

Answer: Thank you for your valuable advice; we have included further discussions based on your advice. Page 21, line 620-624

Q18. Discussion, Lines 375-377 & 385-387. Remove sentences beginning with "Further ...". Not essential.

Answer: Thank you. We have deleted these sentences in revised edition.

Reviewer #2 (Comments for the Author):

Manuscript ID: Spectrum03432-23

Title: Genetic characteristics of human parainfluenza 1 viruses 1-4 associated with acute lower respiratory tract infection in Chinese children, during 2015-2021

This study identified the four types of human parainfluenza viruses (HPIV) in a cohort of children affected with acute lower respiratory tract infection and characterized these strains based on haemagglutinin-neuraminidase (HN) and whole genome sequences. The author has reported the circulation of multiple strains of HPIV1-4 in the population with the emergence of novel B4 substrain of HPIV1, highlighting the significance of continuous surveillance of HPIV. The strength of this article is that it uses a significant number of clinical samples to investigate the molecular diversity and evolutionary pattern of HPIV by performing whole genome sequencing, which is difficult to manage, especially isolated from children.

Q1. However, the methods are poorly written and not described adequately. Similarly, the few

sections of the results (see details in comments) are unclear and elaborated superficially. I would suggest updating this article thoroughly, especially the methods and results.

Please find the detailed concerns listed below.

Answer: Thank you for your valuable feedback. We have extensively revised the entire manuscript, which includes updating the reference sequences obtained from GenBank as of December 31, 2023. We have also recalculated the genetic distances among the genotypes of HPIV1-4 and estimated evolutionary rates. Our goal is to conduct a comprehensive analysis of the results and present readers with more detailed information to contribute useful insights.

Q2. Abstract is missing in the actual text.

Answer: Thanks. The abstract has been included in the revised edition. Page 3, line 61-80.

Q3. Introduction: The author has cited the below statement with reference 7 . I don't see reference number 7 in the list. Please include the reference in the right format in the text like other references instead of showing it in numbers. Notably, the burden of HPIVs in children is comparable to that of RSV infection, although the incidence of ALRTI among different age groups varies widely among populations [7].

Answer: Thank you for your suggestion. The introduction has been thoroughly revised, and references have been included to align with the requirements of Microbiology Spectrum. Page 4-5, line 113-123.

Q4. There is no clarity in the methodology section. How was the library constructed and sequencing performed? Please briefly describe the sequencing method.

Answer: Thanks for your suggestions. All the HPIV-positive samples were further confirmed using a Real-time reverse-transcription polymerase chain reaction kit for HPIV1-4 (Bio-germ, Shanghai, China). Subsequently, samples with a Ct value of less than 28 were sent to the Shanghai BioGerm Medical Technology Limited Company for NGS. The extracted nucleic acid of the HPIV positive samples was captured using the HPIV genome enrichment kit of Shanghai Bio-germ Biotechnology Co., Ltd. The amplified PCR product was purified and quantified, then used an Illumina Nextera XT Kit for deep sequencing on the illumina MiSeq. The sequencing results were analyzed by the CLC Genomics Workbench 12 (Qiagen, Germany). More than 90%

of the sequencing reads reached Q30 (99.9% base call accuracy). The sequencing data volume of each sample was 1Gb with 22-33 million reads. Sequencing depth of over 8000x were carried out for Mapping to reference, with a comparison rate of over 99.99%.

In the revised manuscript, we have incorporated a detailed account of the library construction and sequencing methods. Page 6, line 152-164.

Q5. How many and what reference strains were selected for each type of HPIV for mutational and phylogenetic analysis? Please include the details about reference strains used for comparative nucleotide analysis and for /phylogeny.

Answer: Thank you. In order to provide a comprehensive description of the reference sequences used in this study, we have incorporated a Dataset section into the revised methods of the manuscript. This section provides detailed information on the number of HPIV1-4 sequences retrieved from GenBank, the criteria used for sequence selection, and the final number of sequences included in the analysis. Additionally, all reference sequences utilized in subsequent analyses such as sequence alignment, phylogenetic tree construction, evolutionary rate estimation, variation analysis, recombination analysis, and selection pressure analysis are presented in Supplementary Table 2. Genotype information of the reference strains is listed in Supplementary Tables 3-6.

Q6. How the complete genome of HPIV was assembled. Please describe briefly in the methods. What parameters were considered for interpreting the recombinant events in boot scanning and Simplot analysis? Please include this information in the methods.

Answer: Thank you for your suggestions. All HPIV-positive samples underwent next-generation sequencing (NGS) performed by a sequencing company. The sequencing and sequence assembly were carried out entirely by the company. A concise description of the assembly method is as follows: Each sample's sequencing data volume was 1Gb with 22-33 million reads. Sequencing depth exceeding 8000x was achieved for mapping to reference sequences (NC_001796.2 for

HPIV1, NC_003443.1 for HPIV2, NC_003461.1 for HPIV3, EU627591 for HPIV4b), with a comparison rate exceeding 99.99%. Refer to Page 6, line 152-164.

Specific parameters for recombination analysis have also been included in the revised manuscript. The similarity and bootscanning analyses utilized a sliding window size of 200 bp and a moving step size of 20 bp in Simplot software. Refer to Page 8, line 226-229.

Q7. It is mentioned in the results that All the HPIV1 strains could be categorized into three distinct clades (clade A-C), with the subdivision of clade B into four sub-clades. In what reference, study strains were categorized into clades or subclades? No details are provided in the text.

Answer: Thank you for your revision suggestions. We have supplemented the reference for HPIV2 categorization in the results section to provide audiences with a clearer understanding of HPIV2 classification.(supplementary table 1 and 2)

Q8. It is unclear from the text whether sequence similarity analysis was performed on the full-length genome or only for the HN coding sequence.

Answer: Thank you for your suggestions. We have conducted an analysis of the nucleotide identity among the whole genome sequences obtained from HPIV1-4 in this study, as well as the nucleotide similarity between these strains and their respective prototype strains. This section has been revised accordingly. Please refer to the revised section on page 8, line 222.

Q9. The common amino acid substitution on the HN protein is shown only for HPIV1 & HPIV3 but not for HPIV2 & HPIV4. Why?

Answer: Thank you for your suggestion. In the revised manuscript, we have presented the amino acid mutations of the 156 HPIV1-4 HN protein sequences obtained in this study in table 2, according to consistent mutations within clade, lineage, or sub-lineage.

Q10. The author reported that only the HPIV 1 strain showed a recombination event in the F gene, confirmed by Simplot/bootscan analysis in figure3. However, it is not very clear in figure 3, how the recombination event was determined. Please elaborate figure in the result section and also highlight the recombination region in the figure.

Answer: Thank you for your revisions. In this study, a recombinant sequence was identified at the F gene location (between nucleotide position 4497 and 5543), and we have provided a detailed description of the recombination event in the revised manuscript. Additionally, the recombinant region has been highlighted in red lines in Figure 3 for clarity.

Q11. What are the PSS and NSSs in the table/ and what is the significance of considering these parameters? Please explain these parameters in methods/results. Please expand the abbreviated form in the legend section of the table.

Answer: Thank you for your suggestions. The calculation and quantification of evolutionary pressures represent an essential component of sequence analysis, contributing to our understanding of genetic variation. Several popular statistical methods are utilized for identifying rapidly evolving and unusually conserved sites within protein-coding sequences. These methods rely on estimating site-specific synonymous (dS) and non-synonymous (dN) substitution rate parameters, followed by statistical tests to assess whether $dS \neq dN$. Datamonkey serves as a web-based gateway to a suite of these algorithms, leveraging HyPhy, a molecular evolution analysis platform.

We revised the manuscript to clarify this section. Page 8-9, line 232-240.

[1] Sergei L. Kosakovsky Pond, Simon D. W. Frost, Datamonkey: rapid detection of selective pressure on individual sites of codon alignments, *Bioinformatics*, Volume 21, Issue 10, May 2005, Pages 2531–2533, <https://doi.org/10.1093/bioinformatics/bti320>.

[2] Yang, Z. Inference of selection from multiple species alignments. *Curr. Opin. Genet. Develop.* 2002, 12: 688–694.

[3] Kosakovsky Pond, S.L., et al. HyPhy: hypothesis testing using phylogenies. *Bioinformatics*, 2004,21:676–679.

Q12. I wonder why this study has not estimated the evolutionary rate of all HPIV types based on HF and the complete genome, which is an important parameter if someone is investigating strain diversity in context to time. Please include this analysis.

Answer: Thank you for your valuable suggestion. We have reanalyzed the sequences and calculated the evolutionary rates and 95% highest posterior density (HPD) of HN CDS sequences for each HPIV type. This information has been incorporated into the revised manuscript to provide more useful information for audiences. Page 7, line 186-215 for methods, Page 12, line 339-351 for results and Page 18, line 535-554 for discussion.

Q13. Figure 6 shows two amino acid mutations, N461D and K500R determined in HPIV3 strains, which corresponds to the neutralizing mAb sites in the HN protein. How these mutations were mapped on the prototype strain is not mentioned anywhere in the text. Please include this information in the text.

Answer: Coelingh et al. reported the presence of 13 murine monoclonal antibody binding sites on the HN protein of HPIV3 (corresponding to amino acid residues in the HN protein of prototype strain JN089924 HPIV3/USA/Washington/1957: 171K, 278S, 281A, 345N, 346E, 347N, 364N, 369S, 370P, 395K, 397W, 461N, and 500K). These monoclonal antibody binding sites are likely associated with the formation of B-cell epitopes on the HN protein. In this study, We found two HPIV3 strains possessed two amino acid substitutions (N461D and K500R) in HN protein of HPIV3 corresponding to two reported neutralization-related sites. Previous studies have indicated that the majority of B cell epitopes within the HN protein of HPIV3 are likely associated with conformational epitopes. Aso et al. identified spatial epitopes in the HN protein of HPIV3 that differed from previously reported linear epitope motifs. Therefore, further research is needed to understand the impact of these mutations on viral evasion of host immunity. Page 16, line 456-461.

[1] van Wyke Coelingh, K.L., Winter, C.C., Jorgensen, E.D., Murphy, B.R. Antigenic and structural properties of the hemagglutinin-neuraminidase glycoprotein of human parainfluenza virus type 3: sequence analysis of variants selected with monoclonal antibodies which inhibit infectivity, hemagglutination, and neuraminidase activities. *J. Virol.* 1987, 61 (5), 1473–1477.

[2] Goya, S., Mitchenko, A.L., Viegas, M., Phylogenetic and molecular analyses of human parainfluenza type 3 virus in Buenos Aires, Argentina, between 2009 and 2013: The emergence of new genetic lineages. *Infect. Genet. Evol.* 2016, 39, 85–91.

Takahashi, M., Nagasawa, K., Saito, K., Maisawa, S., Fujita, K., Murakami, K., Kuroda, M., Ryo, A., Kimura, H.. Detailed genetic analyses of the HN gene in human respirovirus 3 detected in children with acute respiratory illness in the Iwate Prefecture, Japan. *Infect. Genet. Evol.* 2018, 59, 155–162.

Q14. What are the A, B1-B4, and C represent in the supp. table 2? Similarly, what do the G1a-G1c, G2, G3, and G4a-G4b represent in supp. table3 and other supp tables? Are these reference strains or the study strains? Please include the statements about whether the evolutionary distances were calculated among study strains or among the prototype strains.

Answer: Thank you for your suggestions. We have We have calculated the intra-group and inter-group genetic distances (p-distances) of each clade, lineage and sub-lineage in the sequence of the database used in Bayesian phylogenetic tree analysis and made a revision in new edition.
Page 6, line 166-178.

Re: Spectrum03432-23R1 (Genetic characteristics of human parainfluenza viruses 1-4 associated with acute lower respiratory tract infection in Chinese children, during 2015-2021)

Dear Prof. Zhengde Xie:

Thank you for the privilege of reviewing your work. Below you will find my comments, instructions from the Spectrum editorial office, and the reviewer comments.

While we are willing to consider a revised version of this paper at Spectrum, it would be in your best interests to improve the writing. I recommend that you ask a colleague of yours who is a native English speaker to read and provide you with some feedback on the writing. You are also welcome to use one of the services here <https://journals.asm.org/content/language-editing-services>

Revision Guidelines

Sincerely,
Biao He
Editor
Microbiology Spectrum

Reviewer #1 (Comments for the Author):

Journal: Microbiology Spectrum

Type of manuscript: Article Revision

Manuscript ID: 03432-23R1

Title: Genetic characteristics of human parainfluenza viruses 1-4 associated with acute lower respiratory tract infection in Chinese children, during 2015-2021

Response to Authors' Revisions:

The authors have satisfactorily addressed this reviewer's first comments.

Reviewer's Additional Comments:

The authors should consider consulting a professional proofreader to help improve text concision and English grammar. Some examples of suggested changes follow:

- 1) Abstract, Line 63. "among young children and the elderly."
- 2) Abstract, Line 71. The authors should review the proper usage of the phylogenetic nomenclature (clade, lineage, sublineage and variant) and use these terms consistently throughout the text.
(https://search.brave.com/search?q=clade+vs+lineage&source=web&summary=1&summary_og=b763e592c5c704bb9e8e15)
- 3) Introduction, Line 117. "ALRTI-related deaths (11). HPIV infections ..."
- 4) Introduction, Line 128-129. "development of vaccines and antivirals."
- 5) Introduction, Line 130. "are limited, especially from China."
- 6) Materials and Methods, Patients and specimens. Instead of on Line 181, this is where to describe the catchment area of the hospital so that the reader can assess how representative this population sample is.
- 7) Materials and Methods, Sequencing. Spell out "NGS" when used for the first time.
- 8) Materials and Methods, Sequencing. What PCR amplification approach was used? Random primers? Multiple over-lapping amplicons?
- 9) Materials and Methods Nucleotide identity, Lines 184-187. Both "identity" and "similarity" are used without distinction.
- 10) Materials and Methods, Line 219. Remove "Furthermore, we"
- 11) Materials and Methods, Line 221. "The rates of molecular evolution of HPIV1-4 were estimated."
- 12) Materials and Methods, Line 241. "The potential positive (PSS) and negative (NSS) selection sites on the predicted HN and F proteins of the HPIV1-3 sequences obtained in this study were evaluated."
- 13) Ethics statement, Lines 270=272. This might be better placed within the Patents and Specimens section.
- 14) Results, Sequence identity analysis, Lines 276-288. The authors should review the proper usage of the terms homology, similarity and identity and use them consistently throughout the text.
- 15) Results, Line 292. "The Bayesian phylogenetic tree constructed from the HN CDS of HPIV1 was categorized into four distinct clades: A, B, C and D."
- 16) Results, Lines 296-297. The sentence, "This finding suggests a localized emergence and spread of these variants within China." is an interpretation of the data and therefore belongs in the Discussion, not the Results.
- 17) Results, Lines 297-298. "of HPIV1 ranged from".

Reviewer #2 (Comments for the Author):

The author addressed all the queries and revised the article adequately by providing detailed datasets. The article has substantially improved and seems suitable for consideration in this journal. Here is a minor suggestion:

The author used the abbreviated form of the coding sequence as "CDS" throughout the text. Please represent it as a complete word in the first place (Entropy plot), then follow the abbreviated form.

Response to Reviewers

Dear editor,

We would like to express our sincere gratitude for the opportunity to revise and resubmit our manuscript [Manuscript ID Spectrum03432-23 and Genetic characteristics of human parainfluenza viruses 1-4 associated with acute lower respiratory tract infection in Chinese children, during 2015-2021]. We greatly appreciate the time and effort that you and the reviewers have invested in providing detailed and constructive feedback.

We have carefully considered the reviewers' comments and have made the necessary revisions to improve the quality and clarity of our manuscript. Specifically, we have addressed all points raised by the reviewers, consulted with a professional proofreading service (American Journal Experts, AJE) to enhance the text concision and English grammar, and ensured consistent usage of technical terms throughout the manuscript.

Enclosed with this letter, you will find a detailed response to each of the reviewers' comments, outlining the changes made to address their suggestions. We believe that these revisions have significantly strengthened our manuscript and hope that it now meets the high standards of your esteemed journal.

Thank you once again for your guidance and support throughout this process. We look forward to your positive response and are hopeful for the acceptance of our revised manuscript.

Sincerely,

Zhengde Xie, M.D. (xiezhengde@bch.com.cn)

Reviewer #1 (Comments for the Author):

Journal: Microbiology Spectrum

Type of manuscript: Article Revision

Manuscript ID: 03432-23R1

Title: Genetic characteristics of human parainfluenza viruses 1-4 associated with acute lower respiratory tract infection in Chinese children, during 2015-2021

Response to Authors' Revisions:

The authors have satisfactorily addressed this reviewer's first comments.

Reviewer's Additional Comments:

The authors should consider consulting a professional proofreader to help improve text concision and English grammar. Some examples of suggested changes follow:

Dear Reviewer,

Thank you for your thorough review and valuable suggestions. We appreciate your effort in improving our manuscript. We would like to inform you that we have utilized the services of the American Journal Experts (AJE) for professional proofreading to enhance the text concision and English grammar throughout our manuscript. Below, we detail the specific actions taken in response to your additional comments:

Q1: Abstract, Line 63. "among young children and the elderly."

Answer: Abstract, Line 33: Revised to "among young children and the elderly."

Q2. Abstract, Line 71. The authors should review the proper usage of the phylogenetic nomenclature (clade, lineage, sublineage and variant) and use these terms consistently throughout the text.

(https://search.brave.com/search?q=clade+vs+lineage&source=web&summary=1&summary_og=b763e592c5c704bb9e8e15)

Answer: Thank you for pointing out the issue regarding the usage of phylogenetic nomenclature

(clade, lineage and sublineage) in our manuscript. We have reviewed and revised the manuscript based on the reference you provided to ensure consistent and proper usage of these terms throughout the text.

Q3. Introduction, Line 117. "ALRTI-related deaths (11). HPIV infections ..."

Answer: Introduction, Line 89: Revised to "ALRTI-related deaths globally (11). HPIV infections ..."

Q4. Introduction, Line 128-129. "development of vaccines and antivirals."

Answer: Introduction, Line 102: Revised to "development of vaccines and antiviral agents."

Q5. Introduction, Line 130. "are limited, especially from China."

Answer: Introduction, Line 102-103: Revised to "are limited, especially from China."

Q6. Materials and Methods, Patients and specimens. Instead of on Line 181, this is where to describe the catchment area of the hospital so that the reader can assess how representative this population sample is. the predicted HN and F proteins of the HPIV1-3 sequences obtained in this study were evaluated."

Answer: Materials and Methods, Patients and Specimens, line 113-119: We have included a description of the catchment area of the hospital in this section to allow readers to assess the representativeness of the population sample.

Q7. Materials and Methods, Sequencing. Spell out "NGS" when used for the first time.

Answer: Materials and Methods, Sequencing, line 141-142: "NGS" has been spelled out as "Next-Generation Sequencing" when used for the first time.

Q8. Materials and Methods, Sequencing. What PCR amplification approach was used? Random primers? Multiple over-lapping amplicons?

Answer: Materials and Methods, Sequencing, Line 114-116: A series of overlapping PCR amplicons were generated using primers designed by BioGerm to efficiently sequence viral genomes directly from clinical specimens.

Q9. Materials and Methods Nucleotide identity, Lines 184-187. Both "identity" and "similarity" are used without distinction.

Answer: Materials and Methods, Nucleotide Identity, Lines 173-177: We have reviewed and ensured the consistent use of "identity" and "similarity" throughout the text.

Q10. Materials and Methods, Line 219. Remove "Furthermore, we"

Answer: Materials and Methods, Line 210: Removed "Furthermore, we".

Q11. Materials and Methods, Line 221. "The rates of molecular evolution of HPIV1-4 were estimated."

Answer: Materials and Methods, Line 212: Revised to "The rates of molecular evolution of HPIV1-4 were estimated."

Q12. Materials and Methods, Line 241. "The potential positive (PSS) and negative (NSS) selection sites on

Answer: Materials and Methods, Line 231-233: Revised to "The potential positive (PSS) and negative (NSS) selection sites on the predicted HN and F proteins of the HPIV1-3 sequences obtained in this study were evaluated."

Q13. Ethics statement, Lines 270-272. This might be better placed within the Patents and Specimens section.

Answer: Ethics Statement, Lines 126-128: This section has been moved within the Patients and Specimens section for better integration.

Q14. Results, Sequence identity analysis, Lines 276-288. The authors should review the proper usage of the terms homology, similarity and identity and use them consistently throughout the text.

Answer: Results, Sequence Identity Analysis, Lines 263-276: We have reviewed and ensured the consistent use of the terms identity and similarity throughout the text.

Q15. Results, Line 292. "The Bayesian phylogenetic tree constructed from the HN CDS of HPIV1 was categorized into four distinct clades: A, B, C and D."

Answer: Results, Line 281-282: Revised to "The Bayesian phylogenetic tree constructed from the HN CDS of HPIV1 was categorized into four distinct clades: A, B, C, and D."

Q16. Results, Lines 296-297. The sentence, "This finding suggests a localized emergence and spread of these variants within China." is an interpretation of the data and therefore belongs in the Discussion, not the Results.

Answer: Results, Lines 285-286: The sentence "This finding suggests a localized emergence and spread of these variants within China." has been moved to the Discussion section as it is an interpretation of the data. Lines 497-500

Q17. Results, Lines 297-298. "of HPIV1 ranged from".

Answer: Results, Lines 286: Revised to "of HPIV1 ranged from".

We hope these revisions address your concerns satisfactorily. Thank you again for your constructive feedback and for helping us improve our manuscript.

Reviewer #2 (Comments for the Author):

The author addressed all the queries and revised the article adequately by providing detailed datasets. The article has substantially improved and seems suitable for consideration in this journal. Here is a minor suggestion:

The author used the abbreviated form of the coding sequence as "CDS" throughout the text. Please represent it as a complete word in the first place (Entropy plot), then follow the abbreviated form.

Dear Reviewer,

Thank you for your positive feedback and for acknowledging the improvements made to our manuscript. We are pleased to hear that the article has substantially improved and is considered suitable for consideration in your esteemed journal.

We appreciate your minor suggestion regarding the use of the abbreviated form "CDS" for the coding sequence. We have revised the manuscript to spell out "coding sequence (CDS)" in its first occurrence, specifically in the Entropy plot section, and then followed it with the abbreviated form throughout the rest of the text.

We hope these changes meet your expectations and further enhance the clarity and readability of our manuscript. Thank you again for your valuable suggestions and for facilitating the review process.

Re: Spectrum03432-23R2 (Genetic characteristics of human parainfluenza viruses 1-4 associated with acute lower respiratory tract infection in Chinese children, during 2015-2021)

Dear Prof. Zhengde Xie:

Your manuscript has been accepted, and I am forwarding it to the ASM production staff for publication. Your paper will first be checked to make sure all elements meet the technical requirements. ASM staff will contact you if anything needs to be revised before copyediting and production can begin. Otherwise, you will be notified when your proofs are ready to be viewed.

Sincerely,
Biao He
Editor
Microbiology Spectrum